# Posterior Sampling via Langevin Monte Carlo for Offline Reinforcement Learning

## Abstract

In this paper, we consider offline reinforcement learning (RL) problems. Within this setting, posterior sampling has been rarely used, perhaps partly due to its explorative nature. The only work using posterior sampling for offline RL that we are aware of is the model-based posterior sampling of [US21]. However, this framework does not permit any tractable algorithm (not even in the linear models) where simulations of posterior samples become challenging, especially in high dimensions. In addition, the algorithm only admits a weak form of guarantees – Bayesian sub-optimality bounds which depend on the prior distribution. To address these problems, we propose and analyze the use of Markov Chain Monte Carlo methods for offline RL. We show that for low-rank Markov decision processes (MDPs), using the Langevin Monte Carlo (LMC) algorithm, our algorithm obtains the (frequentist) sub-optimality bound that competes against any comparator policy $\pi$ and interpolates between $\tilde{\mathcal{O}}(H^2 d\sqrt{C_\pi/K})$ and $\tilde{\mathcal{O}}(H^2\sqrt{dC_\pi/K})$, where $C_\pi$ is the concentrability coefficient of $\pi$, $d$ is the dimension of the linear feature, $H$ is the episode length, and $K$ is the number of episodes in the offline data. For general MDPs with overparameterized neural network function approximation, we show that our LMC-based algorithm obtains the sub-optimality bounds of $\tilde{\mathcal{O}}(H^{2.5}\tilde{d}\sqrt{C_\pi/K})$, where $\tilde{d}$ is the effective dimension of the neural network. Finally, we collaborate our findings with numerical evaluations to demonstrate that LMC-based algorithms could be both efficient and competitive for offline RL in high dimensions.

## 1 Introduction

Offline reinforcement learning (RL) [LGR12] is an important paradigm of RL that finds vast applications in a number of critical domains where online experimentation is costly or dangerous such as healthcare [GJK+19; NBW21], econometrics [KT18; AW21], and robotics [LKTF20]. Formally, a learner is presented with a fixed dataset of tuples of {current state, action, reward, next state} collected by some (unknown) behavior policy during prior interaction with the environment. The goal of offline RL is then to learn a "good" policy out of the offline data without any online interaction. The main challenge toward this goal, due to the lack of exploration capability, is the distribution shift issue, where the distribution induced by the offline data is different from that induced by a comparator policy that the learner wants to compete against.

The approaches to addressing offline RL can be broadly divided into two categories. Maximum likelihood methods with pessimistic adjustments induce *pessimism* by keeping the model close to those supported by the offline data. These were adopted in various forms such as lower-confidence bounds (LCB) [JYW21; RZM+21], version space [US21; XCJ+21], pessimistic regularization [XCJ+21] and primal-dual methods [CJ22; ZHH+22; RZY+22; OPZZ23]. The second approach is based on the Bayesian perspective that relies on samples drawn from a posterior distribution of a statistical model of a target quantity [US21]. This is called posterior sampling (i.e. Thompson sampling). Both of these approaches leverage the idea (either explicitly or implicitly) that more uncertain estimates should be more conservative as the offline data is static. In addition, these approaches complement each other and provide important theoretical guarantees. However, while maximum likelihood methods have been studied extensively for offline RL, posterior sampling has been rarely used for offline RL, perhaps partly due to its explorative nature. Note that this is in stark contrast to online RL settings where posterior sampling has been used and analyzed extensively, e.g.,

see [RVR14; AJ17; Zha22; DMZZ21; ZXZ+22; AZ22]. Indeed, the only work using posterior sampling for offline RL that we are aware of is the model-based posterior sampling of [US21].

However, there are several defects with the current posterior sampling framework for offline RL, in particular, that of [US21]. First, the model-based posterior sampling of [US21] does not permit any tractable algorithms, not even in the linear case. In addition, [US21] only provide Bayesian sub-optimality bounds (averages over a known prior distribution of the model) which are a weaker guarantee than frequentist (i.e., worst-case) sub-optimality bounds. The objective of our paper is precisely to close this gap. To address this problem, we instead consider a value-based posterior sampling framework for offline RL and propose the use of Langevin Monte Carlo (LMC) [WT11] to simulate the samples from the target posterior distribution. While LMC has also been used in online RL settings [MPM+20; XZM+22; HZD23], to the best of our knowledge, the present work is the first to study LMC for offline RL. Our goal is then to propose tractable algorithms and understand the theoretical aspects of using posterior sampling via LMC for offline RL.

**Summary of Contributions and Results.**    We summarize our contributions and results as follows.

- We introduce PPS (pessimistic posterior sampling) – a generic (value-based) posterior sampling framework for offline RL, where we explicitly incorporate pessimism into posterior sampling by simply taking multiple posterior samples and acting pessimistically according to them. Since posterior sampling in our framework is in general still intractable, we propose to use Langevin Monte Carlo (LMC) to draw approximate posterior samples, thus the LMC-PPS framework.

- For linear (low-rank) MDPs, we show that LMC-PPS obtains the sub-optimality bound that interpolates between the worst-case scenario of $\tilde{\mathcal{O}}(H^2 d\sqrt{C_\pi/K})$ and the best-case scenario of $\tilde{\mathcal{O}}(H^2\sqrt{dC_\pi/K})$ (depending on the empirical data), where $C_\pi$ is the concentrability coefficient of any comparator policy $\pi$, $d$ is the dimension of the linear feature, $H$ is the episode length, and $K$ is the number of episodes in the offline data. As a concrete comparison, in tabular MDPs where $d = SA$ with $S$ being the state space cardinality, and $A$ the number of actions, the bound in the best-case scenario above nearly matches the lower bound $\Omega(\sqrt{\frac{H^3 S \min\{C_\pi, A\}}{K}})$ [XJW+21] (up to a gap of $\mathcal{O}(\sqrt{H})$ due to the variance-agnostic nature of the current algorithm). However, there is still a gap of $\mathcal{O}(\sqrt{dH})$ for the worst-case scenario bounds that we left as an open problem.

- To showcase the applicability of the LMC-PPS framework, we also consider a more practical setting of non-linear MDPs where we use neural networks as value function approximation. Different from the exact posterior sampling, the use of LMC in this context adds complications that can compromise theoretical guarantees. We address this problem with a novel algorithmic design using auxiliary linear models. We show that, under standard assumptions, if the network is over-parameterized, LMC-PPS achieves a bound of $H^{2.5}\tilde{d}\sqrt{C_\pi/K}$, where $\tilde{d}$ is the effective dimension of the neural network. This improves the bound of [NTA23] by a factor of $\sqrt{C_\pi}$, due to our more refined analysis.

- In addition, we corroborate our theoretical results with empirical evaluations showing that LMC-PPS could be both efficient and competitive for offline RL.

## 2 PRELIMINARIES

### 2.1 EPISODIC TIME-INHOMOGENOUS MARKOV DECISION PROCESSES

We consider an episodic time-inhomogeneous Markov decision process (MDP) $M = (\mathcal{S}, \mathcal{A}, P, r, H, d_1)$, where $\mathcal{S}$ is the state space, $\mathcal{A}$ is the action space, $P = \{P_h\}_{h \in [H]} \in \{\mathcal{S} \times \mathcal{A} \to \Delta(\mathcal{S})\}^H$ are the transition probabilities [1], $r = \{r_h\}_{h \in [H]} \in \{\mathcal{S} \times \mathcal{A} \to [0,1]\}^H$ is the mean reward functions, $H \in \mathbb{N}$ is the horizon and $d_1 \in \{\mathcal{S} \times \mathcal{A} \to [0,1]\}$ is the initial state distribution. For simplicity, we assume that $|\mathcal{S}|$ is finite but could be exponentially large. [2] A policy $\pi = \{\pi_h\}_{h=1}^H \in \Pi^{all} := \{\mathcal{S} \to \Delta(A)\}^H$ maps each state to a distribution over the action space at each $h$. The state value function $V_h^\pi \in \mathbb{R}^{\mathcal{S}}$ and the action-state value function $Q_h^\pi \in \mathbb{R}^{\mathcal{S} \times \mathcal{A}}$ at each $h$

---

[1] Here $\Delta(\mathcal{X})$ denotes the set of probabilities over $\mathcal{X}$ and $[n] := \{1, \ldots, n\}$.

[2] As large as the total number of atoms in the observable universe $10^{82}$.

are defined as $Q_h^\pi(s,a) = \mathbb{E}_\pi[\sum_{t=h}^H r_t|s_h = s, a_h = a]$, and $V_h^\pi(s) = \mathbb{E}_{a \sim \pi(\cdot|s)}[Q_h^\pi(s,a)]$, where the expectation $\mathbb{E}_\pi$ is taken w.r.t. the randomness of the trajectory induced by $\pi$. For any $V : \mathcal{S} \to \mathbb{R}$, we define the Bellman operator at timestep $h$ as $(\mathbb{B}_h V)(s,a) := r_h(s,a) + (\mathbb{P}_h V)(s,a)$, where $\mathbb{P}_h$ is the transition operator defined as $(\mathbb{P}_h V)(s,a) := \mathbb{E}_{s' \sim P_h(\cdot|s,a)}[V(s')]$. We define an optimal policy $\pi^*$ as any policy that yields the optimal value function, i.e. $V_h^{\pi^*}(s) = \sup_\pi V_h^\pi(s)$ for any $(s,h) \in \mathcal{S} \times [H]$. The occupancy density $d_h^\pi(s,a) := \Pr((s_h,a_h) = (s,a)|\pi)$ is the probability that $(s,a)$ is reached by policy $\pi$ at timestep $h$.

**Data collection.** Consider a fixed dataset $\mathcal{D} = \{(s_h^t, a_h^t, r_h^t, s_{h+1}^t)\}_{h \in [H], t \in [K]}$ generated a priori by some unknown behaviour policy $\{\mu_h^t\}_{h \in [H], t \in [K]}$, where $K$ is the total number of trajectories, $a_h^t \sim \mu_h^t(\cdot|s_h^t), s_{h+1}^t \sim P_h(\cdot|s_h^t, a_h^t)$ for any $(t,h) \in [K] \times [H]$. Define $\mu = \frac{1}{K}\sum_{k=1}^K \mu^k$ and $d_h^\mu = \frac{1}{K}\sum_{k=1}^K d_h^{\mu^k}$. Here $\mu^t$ is allowed to depend no $\{(s_h^i, a_h^i, r_h^i)\}_{h \in [H]}^{i \in [t-1]}, \forall t \in [K]$. This setting of adaptively collected data covers a common practice where the offline data is collected by using some adaptive experimentation [ZRAZ21]. [3]

The value suboptimality of policy $\pi \in \Pi^{all}$ is defined as:
$$\text{SubOpt}_\pi(\hat{\pi}) := \mathbb{E}_{s_1 \sim d_1}[\text{SubOpt}_\pi(\hat{\pi}; s_1)], \text{ where } \text{SubOpt}_\pi(\hat{\pi}; s_1) := V_1^\pi(s_1) - V_1^{\hat{\pi}}(s_1).$$

**Additional Notations.** For any square real-valued matrix $A$, $\lambda_{\min}(A)$ and $\lambda_{\max}(A)$ denotes the smallest and largest eigenvalue of $A$, respectively. Denote $\kappa(A)$ the condition number of $A$, i.e., $\kappa(A) = \frac{\lambda_{\max}(A)}{\lambda_{\min}(A)}$. Let $\|x\|_A := \sqrt{x^T A x}$. Denote $x \lesssim y$ to mean $x = \mathcal{O}(y)$. We write $x = o_m(1)$ to mean that $x = \mathcal{O}(m^{-\alpha})$ for some $\alpha > 0$. Let $\Phi(\cdot)$ be the c.d.f. of the standard normal distribution. Let $\tilde{\mathcal{O}}(\cdot)$ be $\mathcal{O}(\cdot)$ with hidden log factors.

## 2.2 Offline RL with Value Function Approximations

We consider large state space settings where we employ some function classes to estimate the value functions. In particular, in this paper, we consider linear function approximation and (overparameterized) neural network function approximation.

### 2.2.1 Low-rank MDPs

For linear function approximation, we consider low-rank (linear) MDPs which have been extensively studied in offline [JYW21; YDWW22; NTYG+22] and online setting [JYWJ20; AKKS20; MCK+21].

**Definition 1** (Low-rank MDP). *An MDP is said to be low-rank of $d$ if for all $h \in [H]$, there exist a known feature map $\phi_h : \mathcal{S} \times \mathcal{A} \to \mathbb{R}^d$, a reward parameter $w_h \in \mathbb{R}^d$, and a mapping $\rho_h : \mathcal{S} \to \mathbb{R}_+^d$ such that $\|\rho_h(s)\|_1 = 1, \forall s$ such that*
$$r_h(s,a) = \langle \phi_h(s,a), w_h \rangle, \qquad P_h(s'|s,a) = \langle \phi_h(s,a), \rho_h(s') \rangle, \forall (s,a,s',h).$$
*where $\sup_{(s,a)} \|\phi_h(s,a)\|_2 \leq 1$, $\|w_h\|_2 \leq \sqrt{d}$, and $|\int \rho_h(s)v(s)ds| \leq \sqrt{d}\|v\|_\infty$.*

### 2.2.2 General MDPs with Neural Function Approximation

We also consider a more general setting where the underlying MDP does not admit a low-rank structure and we need to resort to a more complex function approximation for learning the value functions. In particular, we consider the setting where we use a ReLU neural network as a function approximator. For simplicity, we denote $\mathcal{X} := \mathcal{S} \times \mathcal{A}$ and view it as a subset of $\mathbb{R}^d$; we also denote $x_h^t = (s_h^t, a_h^t)$ and $x = (s,a)$ throughout the paper. Without loss of generality (w.l.o.g.), we assume $\mathcal{X} \subset \mathbb{S}_{d-1} := \{x \in \mathbb{R}^d : \|x\|_2 = 1\}$. We consider a standard two-layer neural network:
$$f(x;\theta) := f(x;W,b) = \frac{1}{\sqrt{m}}\sum_{i=1}^m b_i \sigma(w_i^T x), \tag{1}$$
where $m$ is an even number, $\sigma(\cdot) = \max\{\cdot, 0\}$ is the ReLU activation function, $b_i \in \mathbb{R} \, \forall i \in \{1, \ldots, m\}$, and $\theta = (w_1^T, \ldots, w_m^T)^T \in \mathbb{R}^{md}$.

---

[3]When $\mu^1 = \cdots = \mu^K$, it recovers the setting of independent episodes in [DJW20].

**Symmetric initialization.** During training, we initialize $(\theta, b)$ via the symmetric initialization scheme [GCL$^+$19] as follows: For any $i \leq \frac{m}{2}$, $w_i = w_{\frac{m}{2}+i} \sim \mathcal{N}(0, I_d/d)$, and $b_{\frac{m}{2}+i} = -b_i \sim$ Unif($\{-1, 1\}$). [4] For simplicity, during training we fix all $b_i$ after initialization and only optimize over $\theta$, thus we write $f(x; \theta, b)$ as $f(x; \theta)$. Let $\mathcal{F}_{nn}(m)$ denote the set of such $f(x; \theta)$. Denote $g(x; \theta) = \nabla_\theta f(x; \theta) \in \mathbb{R}^{md}$, and let $\theta_0$ be the initial parameters of $\theta$. We allow the network to be *overparameterized* in the sense that the width $m$ can be sufficiently larger than the sample size $K$.

Our analysis is motivated by the recent developments in understanding the dynamics of (S)GD in overparameterized neural networks. Given the initialization above, and sufficient parameterization, we can show that the iterates of (S)GD tend to stay close to its initialization; thus the first-order Taylor expansion $f(x; \theta) \approx f(x; \theta_0) + \langle g(x; \theta_0), \theta - \theta_0 \rangle$ can be used as a proxy to track the dynamics of the training weights [ADH$^+$19; AZLS19; HN19; CG19; Bel21]. This behaviour can be captured in the framework of Neural Tangent Kernel (NTK) [JGH18].

**Definition 2** (NTK [JGH18]). *Let $\sigma'(u) = \mathbb{1}\{u \geq 0\}$. The NTK kernel $K_{ntk} : \mathcal{X} \times \mathcal{X} \to \mathbb{R}$ is defined as*

$$K_{ntk}(x, x') = \mathbb{E}_{w \sim \mathcal{N}(0, I_d/d)} \left\langle x\sigma'(w^T x), x'\sigma'(w^T x') \right\rangle.$$

Let $\mathcal{H}_{ntk}$ be the reproducing kernel Hilbert space (RKHS) induced by $K_{ntk}$. Note that $K_{ntk}$ is a universal kernel [JTX20], thus $\mathcal{H}_{ntk}$ is dense in the space of continuous functions on $\mathcal{X}$ [RR08].

## 3 POSTERIOR SAMPLING FOR OFFLINE REINFORCEMENT LEARNING

We consider a generic value-based posterior sampling framework for offline RL below. The main challenge for the algorithmic design is how to ensure pessimism in posterior sampling (PS). The only work about posterior sampling for offline RL that we are aware of is the model-based PS of [US21]. A key advantage of the model-based PS of [US21] is that the algorithm does not require explicit pessimism enforcement. That is, pessimism is *implicit* from posterior sampling. However, the implicit pessimism emerges at the cost of sacrificing the *frequentist* (i.e., worst-case) sub-optimality bounds for the *Bayesian* sub-optimality bounds (averages over a known prior distribution).

---

**Algorithm 1** PPS: Pessimistic Posterior Sampling for offline RL

---

**Input:** Offline data $\mathcal{D}$, number of posterior samples $M$, truncation parameter $\iota$
1: Initialize $\tilde{V}_{H+1} \equiv 0$
2: **for** $h = H, \ldots, 1$ **do**
3: $\quad \{\tilde{f}_h^i\}_{i \in [M]} \overset{i.i.d.}{\sim} Pr_h(\cdot | \mathcal{D}, \tilde{V}_{h+1})$ $\qquad\qquad\qquad\qquad$ ▷ *Posterior sampling*
4: $\quad \widetilde{Q}_h(\cdot, \cdot) \leftarrow \min\{\min_{i \in [M]} \tilde{f}_h^i(\cdot, \cdot), (H - h + 1)(1 + \iota)\}^+$ $\qquad\qquad$ ▷ *Pessimism*
5: $\quad \widetilde{\pi}_h \leftarrow \arg\max_{\pi_h \in \Pi} \langle \widetilde{Q}_h, \pi_h \rangle$ and $\widetilde{V}_h(\cdot) \leftarrow \langle \widetilde{Q}_h(\cdot, \cdot), \widetilde{\pi}_h(\cdot | \cdot) \rangle$.
6: **end for**
**Output:** $\tilde{\pi} = (\tilde{\pi}_1, \ldots, \tilde{\pi}_H)$

---

To obtain high probability frequentist bounds for PS, we consider a simple algorithmic framework that naturally incorporates pessimism into posterior sampling, namely Pessimistic Posterior Sampling (PPS). We present the pseudo-code of PPS in Algorithm 1. At each timestep $h \in [H]$, given the offline data $\mathcal{D}$ and the previous value estimate $\tilde{V}_{h+1}$, PPS constructs a posterior distribution $Pr_h(\cdot | \mathcal{D}, \tilde{V}_{h+1})$ over the function space $\{\mathcal{S} \times \mathcal{A} \to \mathbb{R}\}$. To enforce pessimism into posterior sampling, PPS draws several posterior samples and acts pessimistically with respect to them. [5]

However, exact sampling from the data posterior $Pr_h(\cdot | \mathcal{D}, \tilde{V}_{h+1})$ is usually intractable in high dimensions and an MCMC algorithm has to be used. Consequently, the main objective and contribution of our paper is to devise new algorithms and analyses by considering the additional complexity

---

[4]This symmetric initialization scheme makes $f(x; \theta_0) = 0$ and $\langle g(x; \theta_0), \theta_0 \rangle = 0$ for any $x$.

[5]It's worth noting that the idea of taking multiple samples and acting accordingly (either optimistically for online RL and pessimistically for offline RL) in the PPS framework of Algorithm 1 is not new as it has been explored, especially in the context of reward-perturbing algorithms [KZS$^+$20; ICN$^+$21; TBC$^+$22; NTA23]. Thus, it would be best to view the PPS framework above as a natural protocol that we consider to study Langevin Monte Carlo for offline RL.

of using approximate samples of the data posterior. Specifically, we consider Langevin Monte Carlo (LMC) [6], a gradient-based MCMC scheme to draw approximate samples from the data posterior. In spirit, LMC simply performs noisy gradient descent to minimize squared value errors. However, in high dimensions, directly applying LMC can compromise the convergence guarantees. We describe in detail how we use LMC in low-rank MDPs and general MDPs in the following subsections.

## 3.1 LMC FOR LOW-RANK MDPs

We consider low-rank MDPs defined in Definition 1 with known feature maps $\phi_h$. The LMC framework for approximate posterior sampling is presented in Algorithm 2. In particular, given the previous value estimate $\widetilde{V}_{h+1}$ and the offline data $\mathcal{D}$, we construct the regularized squared temporal-difference (TD) loss $L_h$ defined in Line 1. We obtain approximate posterior samples by simply adding controlled Gaussian noise to the gradient update to minimize the loss $L_h$ (Line 4). [7] We dub Lin-LMC-PPS as the combination of Lin-LMC in Algorithm 2 and PPS of Algorithm 1.

---

**Algorithm 2** Lin-LMC

---

**Input:** Offline data $\mathcal{D}$, previous value estimate $\widetilde{V}_{h+1}$, regularization parameter $\lambda$, number of posterior samples $M$, learning rate $\eta$, noise variance $\tau$, iteration number $T$
1: Set $L_h(\theta) := \sum_{k=1}^{K} (\phi_h(s_h^k, a_h^k)^T \theta - r_h^k - \widetilde{V}_{h+1}(s_{h+1}^k))^2 + \lambda\|\theta\|_2^2$ and initialize $\theta_{h,0}^i \equiv 0, \forall i$.
2: **for** $i = 1, \dots, M$ **do**
3:     **for** $t = 1, \dots, T$ **do**
4:         $\theta_{h,t}^i \leftarrow \theta_{h,t-1}^i + \eta\nabla L_h(\theta_{h,t-1}^i) + \sqrt{2\eta\tau}\epsilon_t$ where $\epsilon_t \sim \mathcal{N}(0, I_d)$
5:     **end for**
6:     Set $\tilde{f}_h^i = \langle \phi_h(\cdot, \cdot), \theta_{h,T}^i \rangle$
7: **end for**
**Output:** $\{\tilde{f}_h^i\}_{i \in [M]}$

---

## 3.2 LMC FOR GENERAL MDPs WITH NEURAL FUNCTION APPROXIMATION

We now consider an LMC-based approximate posterior sampling scheme for general MDPs, where the action-value functions are approximated by overparameterized neural networks defined in Equation (1). The pseudo-code for this algorithm is presented in Algorithm 3. Designing an LMC-based scheme for this setting requires more subtlety than for low-rank MDPs. First of all, for each timestep $h$, instead of using the data from all episodes, we only use data from the set of episodes $\mathcal{I}_h := [(H - h)K' + 1, \dots, (H - h + 1)K']$ with $K' := \lfloor K/H \rfloor$. We adopt this data splitting technique from [NTA23] to avoid the complications of the data dependence induced in the neural network setting. We now explain in detail the other components of the algorithm as follows.

It is tempting to directly apply LMC to the squared TD loss $\mathcal{L}_h(\theta) := \sum_{k \in \mathcal{I}_h} (f(x_h^k; \theta) - r_h^k - \widetilde{V}_{h+1}(s_{h+1}^k))^2 + \lambda\|\theta - \theta_{h,0}\|_2^2$, just like Algorithm 2. While, in practice, this may be a natural choice, it poses two main technical difficulties for the theoretical analysis in the overparameterized setting: (i) a uniform convergence argument cannot apply to an overparameterized setting, and (ii) sufficient pessimism moves the network out of the NTK regime. For (i), since $\widetilde{V}_{h+1}$ depends on $(s_h^k, a_h^k)$ if we use up all data for computing $\widetilde{V}_{h+1}$, we typically need to construct an infinity-norm cover of the function class of $\widetilde{V}_{h+1}$ in analyzing the regression $r_h^k + \widetilde{V}_{h+1}(s_{h+1}^k)$ [JYW21]. However, since the class of $\widetilde{V}_{h+1}$ is an overparameterized neural network – whose complexity scales polynomially with the network width $m$, this analysis results in a vacuous bound in the overparameterized setting where $m$ also scales in a high-order polynomial of $K$. For (ii), since we will need to maintain sufficient pessimism, we need the noise level in LMC to be sufficiently large – in particular, $\tau = \Omega(1)$. However, adding that much noise directly to the network weights during the training causes the weights to deviate from the initial value $\theta_0$ as much as $\sqrt{\tau m d}$ (in 2-norm), effectively moving the network dynamics out of the NTK regime where the weights stay close to the initial values.

---

[6] It is possible to use the Metropolis-Adjusted Langevin Algorithm to correct the bias introduced by LMC. As this bias can be controlled in our settings, to avoid any further complications, we stick with LMC to draw approximate posterior samples.

[7] In this case, $Pr_h(\theta|\mathcal{D}, \widetilde{V}_{h+1}) \propto \exp(-L_h(\theta))$.

To overcome this issue, we decouple neural networking training and weight perturbation into two separate phases. The first phase (Line 1-Line 5) simply trains the neural network with standard gradient descent (GD) to minimize the regularized squared TD loss $\mathcal{L}_h$ (defined in Line 2). Given the trained weight $\theta_{h,T_1}$ from the first phase, in the second phase (Line 6-Line 16), we train an auxiliary *linear* model $f_{lin}(\cdot,\cdot;\theta) := \langle g(\cdot,\cdot;\theta_{h,T_1}), \theta \rangle$ on the offline data using the standard gradient descent (Line 8-Line 10) to obtain $\theta_{h,T_2}^{lin}$ and using LMC (Line 12-Line 14) to obtain $\theta_{h,T_2}^{lin,i}$ for each $i \in [N]$. The intuition is to simulate perturbations in the neural network that would have been caused by exact posterior sampling, without actually perturbing the network weights during the training in the first phase. The perturbation $\langle g(\cdot,\cdot;\theta_h), \theta_{h,T_2}^{lin,i} - \theta_h^{lin} \rangle$ is then augmented to the trained network $f(\cdot,\cdot;\theta_{h,T_1})$ to form an approximate posterior sample $\tilde{f}_h^i$ (Line 15). Finally, we dub the combination of PPS and Neural-LMC as Neural-LMC-PPS.

---

**Algorithm 3** Neural-LMC

**Input:** Offline data $\mathcal{D}$, previous value estimate $\tilde{V}_{h+1}$, regularization parameter $\lambda$, number of posterior samples $M$, learning rates $\eta_1, \eta_2$, noise variance $\tau$, iteration numbers $T_1, T_2$

1: **PHASE I: Standard network training with GD**
2: Set $\mathcal{L}_h(\theta) = \sum_{k \in \mathcal{I}_h} (f(x_h^k;\theta) - r_h^k - \tilde{V}_{h+1}(s_{h+1}^k))^2 + \lambda \|\theta - \theta_{h,0}\|_2^2$ and where $\theta_{h,0}$ is initialized using the symmetric scheme
3: **for** $t = 1, \ldots, T_1$ **do**
4: $\quad \theta_{h,t} \leftarrow \theta_{h,t-1} + \eta_1 \nabla \mathcal{L}_h(\theta_{h,t-1})$
5: **end for**
6: **PHASE II: Auxillary linear model training with LMC**
7: Set $\mathcal{L}_h^{lin}(\theta) \leftarrow \sum_{k \in \mathcal{I}_h} (\theta^T g(x_h^k;\theta_{h,T_1}) - r_h^k - \tilde{V}_{h+1}(s_{h+1}^k))^2 + \lambda \|\theta\|_2^2$, and $\theta_{h,0}^{lin,i} = \theta_{h,0}^{lin} \equiv 0, \forall i$
8: **for** $t = 1, \ldots, T_2$ **do**
9: $\quad \theta_{h,t}^{lin} \leftarrow \theta_{h,t-1}^{lin} + \eta \nabla \mathcal{L}_h^{lin}(\theta_{h,t-1}^{lin})$
10: **end for**
11: **for** $i = 1, \ldots, M$ **do**
12: $\quad$ **for** $t = 1, \ldots, T_2$ **do**
13: $\quad\quad \theta_{h,t}^{lin,i} \leftarrow \theta_{h,t-1}^{lin,i} + \eta \nabla \mathcal{L}_h^{lin}(\theta_{h,t-1}^{lin,i}) + \sqrt{2\eta\tau}\epsilon_t$ where $\epsilon_t \sim \mathcal{N}(0, I_{md})$
14: $\quad$ **end for**
15: $\quad \tilde{f}_h^i(\cdot,\cdot) \leftarrow f(\cdot,\cdot;\theta_h) + \langle g(\cdot,\cdot;\theta_{h,T_1}), \theta_{h,T_2}^{lin,i} - \theta_{h,T_2}^{lin} \rangle$
16: **end for**
**Output:** $\{\tilde{f}_h^i\}_{i \in [M]}$

---

## 4 THEORETICAL ANALYSIS OF LMC-PPS FOR OFFLINE RL

In this section, we provide the sub-optimality analysis of (Lin/Neural)-LMC-PPS algorithms. To measure the distribution mismatch between a comparator policy $\pi$ and the behavior policy $\mu$, we adopt the single-policy concentrability coefficient [LSAB19; RZM+21].

**Definition 3** (Single-policy concentrability coefficient). *For any policy $\pi$, the (single-policy) concentrability coefficient $C_\pi$ of $\pi$ is defined by:* $C_\pi := \sup_{h,s,a} \frac{d_h^\pi(s,a)}{d^\mu(s,a)}$.

$C_\pi$ is fully characterized by the MDP, comparator policy $\pi$, and the behavior policy $\mu$. It is weaker than the classic uniform concentrability coefficient [SM05] and has been used extensively in recent offline RL works [YW21; NTGNV22; NTYG+22; JRYW22; NTA23; LSC+22].

### 4.1 GUARANTEES OF LIN-LMC-PPS

**Theorem 1.** *Fix any $\delta > 0$. Consider the low-rank MDP in Definition 1 and let $\tilde{\pi}$ be the policy returned by Lin-LMC-PPS, in which we set $M = \ln \frac{H|\mathcal{S}|}{\delta} / \ln \frac{1}{1-\Phi(-1)}$, $T \geq \frac{-\ln 4}{\ln(1-(2\max_h \kappa(\Lambda_h))^{-1})} \approx \kappa(\Lambda_h)$, $\sqrt{\tau} = \tilde{\mathcal{O}}(H\sqrt{d} \max\{\sqrt{\lambda}, 1\})$, $\eta\lambda < 1/2$, $\iota \geq (1 - 2\eta\lambda)^T HK/\sqrt{\lambda}$, where $\Lambda_h := \lambda I + \sum_{k=1}^K \phi_h(s_h^k, a_h^k)\phi_h(s_h^k, a_h^k)^T$. Then with probability at least $1 - 4H\delta$, for any $\pi \in \Pi^{all}$, we have*

$$\text{SubOpt}_\pi(\tilde{\pi}) \lesssim \sqrt{\frac{C_\pi \tau \ln(KHM/\delta)}{K}} \sum_{h=1}^H \sqrt{\sum_{i=1}^d \frac{\lambda_i}{\lambda + \lambda_i}} + \zeta_{opt} + \zeta_{est},$$

where $\{\lambda_i\}_{i\in[d]}$ are the eigenvalues of $\sum_{k=1}^K \phi_h(s_h^k, a_h^k)\phi_h(s_h^k, a_h^k)^T$, $C_\pi$ is defined in Definition 3, $\zeta_{est} = H^2\sqrt{\frac{C_\pi}{K}\ln\frac{\ln KC_\pi H2}{\delta}} + H\sqrt{\frac{H\ln(1/\delta)}{K}}$, and $\zeta_{opt} = \sqrt{C_\pi}(1-2\eta\lambda)^T \frac{H^2(1+\iota)K}{\sqrt{\lambda}}$.

**Interpretation.** The optimization error $\zeta_{opt}$ can be driven to be arbitrarily small with large $T$, thus not a dominant term. Several remarks are in order. Theorem 1 provides a family of upper bounds on the sub-optimality of the learned policy $\tilde\pi$, indexed by the choice of the comparator policy $\pi \in \Pi^{all}$. This is meaningful in practice as there is no reason to expect that $C_{\pi^*}$ is a small quantity given the offline data is given a prior. Note that the bound holds with high probability and is in the frequentist (worst-case) nature, which is stronger than the expected Bayesian bound of [US21].

**Tight confidence bounds.** An intriguing property is that the confidence parameter $\sqrt{\tau}$ scales in the order of $\sqrt{d}$, instead of the order of $d$ as in the LCB-based methods [JYW21]. [8] This $\sqrt{d}$ improvement comes from the different ways PPS enforces pessimism. In the LCB-based algorithm, pessimism is incorporated by explicitly subtracting a bonus term from the regression estimate. This enlarges the function space from a linear space to a quadratic space. This function enlargement results in a sub-optimal $d$ scaling, due to that the bonus function is data-dependent and we need a uniform convergence argument over this random bonus to leverage the concentration phenomenon. In contrast, posterior samples in Lin-LMC-PPS do not enlarge the function space, thus saving the $\sqrt{d}$ factor from the uniform convergence argument. This tight confidence makes it possible to have a near-optimal bound in some "begnin" regimes, as discussed in the following.

**Interpolating bounds.** The bounds in Theorem 1 is *data-adaptive* in the sense that it depends on the scaling of the eigenvalues $\{\lambda_i\}_{i\in[d]}$ of the unregularized empirical covariance matrix $\sum_{k=1}^K \phi_h(s_h^k, a_h^k)\phi_h(s_h^k, a_h^k)^T$. In particular, the bounds interpolate from the worst-case-scenario bound $\tilde{\mathcal{O}}(H^2 d\sqrt{C_\pi/K})$ to the best-case-scenario bound $\tilde{\mathcal{O}}(H^2\sqrt{dC_\pi/K})$ (when $\sum_{i=1}^d \lambda_i = \tilde{\mathcal{O}}(1)$, which could occur in high dimensions when $d \gg K$). As a concrete comparison, in tabular MDPs, our bounds translate into $\tilde{\mathcal{O}}(H^2 SA\sqrt{C_\pi/K})$ for the worst case and $\tilde{\mathcal{O}}(H^2\sqrt{SAC_\pi/K})$ for the best case, which nearly match the lower bound $\Omega(\sqrt{\frac{H^3 S\min\{C_\pi,A\}}{K}})$ [XJW+21]. [9]

## 4.2 Guarantees of Neural-LMC-PPS

Our sub-optimality bounds depend on a notion of effective dimension.

**Definition 4** (Effective dimension). *For any $h \in [H]$ and some index set $\mathcal{I}_h \subseteq [K]$, the effective dimension of the NTK matrix on data $\{x_h^k\}_{k\in\mathcal{I}_h}$ is defined as*

$$\tilde{d}_h := \frac{\ln\det(I_K + \mathcal{K}_h/\lambda)}{\ln(1 + |\mathcal{I}_h|/\lambda)}, \forall h \in [H], \text{ and } \tilde{d} := \max_{h\in[H]} \tilde{d}_h,$$

*where $\mathcal{K}_h := [K_{ntk}(x_h^i, x_h^j)]_{i,j\in\mathcal{I}_h}$ is the Gram matrix of $K_{ntk}$ on the data $\{x_h^k\}_{k\in\mathcal{I}_h}$.*

The effective dimension is widely used in contextual bandits [SKKS09; VKM+13; ZLG20; CG17] and in RL [YW20; KKL+20]. We define the function class

$$\mathcal{Q}^* := \left\{ x \mapsto \int_{\mathbb{R}^d} c(w)^T x\sigma'(w^T x)dw : c(w) \in \{\mathbb{R}^d \to \mathbb{R}\} \text{ and } \sup_w \frac{\|c(w)\|_2}{p_0(w)} < B \right\}$$

where $B > 0$ is some absolute constant and $p_0$ is the probability density function of $\mathcal{N}(0, I_d/d)$. Note that $\mathcal{Q}^*$ is a dense subset of $\mathcal{H}_{ntk}$ [GCL+19, Lemma C.1] when $B = \infty$. To interpret our results in the present paper, it's convenient to view $B$ as fixed while letting $K$ become large.

We impose an assumption on the closeness of the Bellman operator.

**Assumption 4.1** (Approximate completeness). *There exist $\xi_h \geq 0, \forall h \in [H]$ such that*

$$\sup_{f\in\mathcal{F}_{nn}} \inf_{g\in\mathcal{Q}^*} \|\mathbb{B}_h f - g\|_\infty \leq \xi_h, \forall h \in [H].$$

---

[8] With a refined analysis using the advantage-reference decomposition, [XZS+22] showed that the LCB-based algorithm also needs the confidence parameter in the order of $\sqrt{d}$. However, this relies on an *explorative* assumption, i.e., $\min_{h\in[H]} \lambda_{\min}\left(\mathbb{E}_{(s_h,a_h)\sim d_h^\mu}[\phi_h(s_h,a_h)\phi_h(s_h,a_h)^T]\right) > 0$, which we do not require.

[9] The gap $\mathcal{O}(\sqrt{H})$ could potentially be closed by using the variance-weighted regression in [YDWW22; XZS+22] which could be added to our current algorithms.

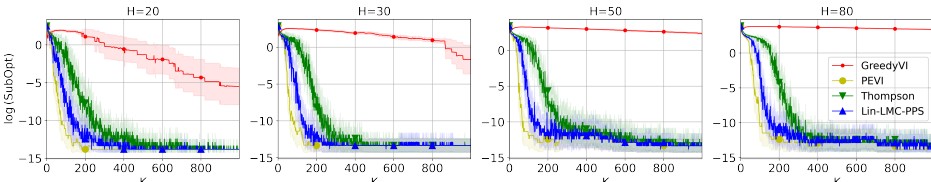

Figure 1: Value suboptimality (in log scale) in a hard instance of linear MDP as a function of sample size $K$ for episode lengths $H \in \{20, 30, 50, 80\}$. Four methods: GreedyVI, PEVI, Thompson sampling, and Lin-LMC-PPS (ours) are presented.

Assumption 4.1 ensures that $\mathbb{B}_h$ applied in the class of overparameterized neural networks can be captured by infinite-width neural network up to some error $\xi$. This is a common assumption in the literature of RL with neural function approximation [CYLW19; WSY20; YJW+20; YWW22].

Define $\Lambda_h := \lambda I + \sum_{k \in \mathcal{I}_h} g(x_h^k; \theta_h) g(x_h^k; \theta_h)^T$ and $\bar{\Lambda}_h := \lambda I + \sum_{k \in \mathcal{I}_h} g(x_h^k; \theta_0) g(x_h^k; \theta_0)^T$.

**Theorem 2.** *Fix any $\delta > 0$. Let $\tilde{\pi}$ be the policy returned by Neural-LMC-PPS, in which we set $m \gtrsim poly(K', H, d, B, \lambda, 1/\delta)$, $\iota \gtrsim (1 - 2\eta\lambda_{\max}(\bar{\Lambda}_h))^{T_1} \lambda^{-1/2} K' H (1 + \iota) + o_m(1)$, $\eta_1 \leq \frac{1}{2\lambda_{\max}(\bar{\Lambda}_h)}$, $\eta_2 = \frac{1}{4\lambda_{\max}(\Lambda_h)}$, $T_2 \geq \frac{\ln 2}{\ln(1 + \frac{1}{2\kappa(\Lambda_h) - 1})}$, $M = \ln \frac{H|\mathcal{S}|}{\delta} / \ln \frac{1}{1 - \Phi(-1)}$, $\lambda = 1 + \frac{1}{K'}$, and $\sqrt{\tau} = 4H\sqrt{\tilde{d} \ln(1 + K'/\lambda) + 1 + 2\ln(3H/\delta)} + B\sqrt{\lambda} + \xi\sqrt{K'}/\lambda + o_m(1)$. Then, under Assumption 4.1, with probability at least $1 - 4\delta$ over the randomness of the offline data and network initialization, for any $\pi \in \Pi^{all}$ such that $C_\pi < \infty$, we have*

$$\text{SubOpt}_\pi(\tilde{\pi}) \lesssim H\sqrt{\frac{\tau \tilde{d} C_\pi}{K'} \ln \frac{MK'H}{\delta} \ln(1 + \frac{K'}{\delta})} + \zeta_{opt} + \zeta_{msp} + \zeta_{est},$$

*where $\zeta_{opt} = H(1 + \iota)\sqrt{\frac{C_\pi K'}{\lambda}} \sum_{h=1}^{H} (1 - 2\eta_1 \lambda_{\min}(\bar{\Lambda}_h))^{T_1}$, $\zeta_{msp} = \sqrt{\frac{C_\pi}{K'}} \sum_{h=1}^{H} \xi_h + o_m(1)$, and $\zeta_{est} = \frac{H(1 + \iota)}{\sqrt{K'}}(H\sqrt{C_\pi \ln \ln \frac{4K' C_\pi H^2}{\delta}} + \sqrt{2} + 2\sqrt{H \ln \frac{1}{\delta}})$.*

The bound consists of four main terms. The first term is the excess error and is of main interest on account of being the dominating term. The second term $\zeta_{opt}$ is the optimization error of gradient descent which can be made arbitrarily small with large $T_1$. The third term stems from the misspecification in the representational assumption of the Bellman operator (Assumption 4.1) and the approximation error of a neural network to its first-order Taylor expansion in the NTK regime. The last term $\zeta_{est}$ is a low-order estimation error. If we assume that there is no misspecification error, i.e., $\xi_h = 0, \forall h \in [H]$, then $\text{SubOpt}_\pi(\tilde{\pi}) = \tilde{\mathcal{O}}(H^2 \tilde{d} \sqrt{C_\pi/K'})$. Note that our upper bounds do not scale with the (neural tangent) feature dimension $md$, which is polynomial in the sample size $K$. Instead, our bound scales polynomially with the effective dimension $\tilde{d}$, which is a data-dependent quantity. This is in contrast with the bounds of Langevin-type algorithms for (online) bandits with finite-dimensional parametric function classes, which scale polynomially with feature dimension [MPM+20; XZM+22]. Compared to the bound of [NTA23] that also considers offline RL in general MDPs with overparameterized neural network function approximation, our bound improves by a factor of $\sqrt{C_\pi}$ due to a more refined analysis.

## 5 EXPERIMENTS

In this section, we empirically evaluate the efficacy of LMC for offline RL. We compare the proposed algorithm against the following approaches: (i) Greedy methods (GreedyVI and NeuralGreedy for linear and neural cases, respectively) which fit a given model to the data without any perturbation or uncertainty quantification; (ii) LCB-based methods, namely LinLCB [JYW21] and NeuraLCB [NTGNV22] that construct LCB using linear and neural tangent features, respectively; and (iii) exact Thompson sampling (we refer to it as Thompson and NeuralTS in the linear and neural cases, respectively) that directly performs posterior sampling from an exact posterior. For Neural-LMC-PPS, we directly apply noisy gradient updates to the network, instead of using an auxiliary linear model, to approximate posterior samples (see the supplementary for details).[10]

---

[10]In our implementation, we do not grow the network width (polynomially) with data size. Nonetheless, our method exhibits fundamental aspects of approximate posterior sampling via LMC.

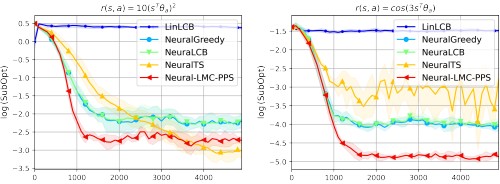
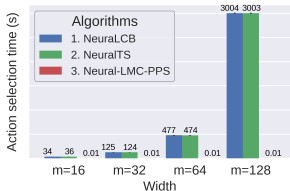

Figure 2: Value suboptimality (in log scale) as a function of sample size $K$ in non-linear contextual bandits with the reward functions: (left) $r(s,a) = 10(s^T\theta_a)^2$, (right) $r(s,a) = \cos(3s^T\theta_a)$.

Figure 3: Action selection run-time (in seconds) of NeuraLCB, NeuralTS, and Neural-LMC-PPS.

We evaluated all methods in two settings: (1) a hard instance of linear MDP [JYWJ20], and (2) non-linear contextual bandits with (a) $r(s,a) = 10(s^T\theta_a)^2$, and (b) $r(s,a) = \cos(3s^T\theta_a)^2$, where $s, \theta_a \sim Unif(\mathbb{S}_{d-1})$ with $d = 16$ and $|\mathcal{A}| = 10$ (see the supplementary for details).

In Figure 1, the greedy methods fail with partial data coverage which reaffirms the practical benefits of pessimism for offline RL. We also see that Lin-LMC-PPS nearly matches the performance of PEVI [JYW21] and even slightly outperforms the exact posterior sampling method (Thompson). In Figure Figure 2, we observe the benefit of neural representation in non-linear problems where methods using linear models clearly fail. Again, we observe that Neural-LMC-PPS offers a competitive performance compared to LCB-based and exact posterior sampling methods in this case, even without computing any LCB or data posterior. In Figure 3, we show that Neural-LMC-PPS enjoys constant action selection time while NeuraLCB and NeuralTS grow in time with the network width (due to computing LCB and an exact posterior computation) for action selection.

## 6 CONCLUSION AND DISCUSSION

**Conclusion.** We presented the first framework for posterior sampling via Langevin Monte Carlo for offline RL. We showed that the framework is provably sample-efficient in low-rank MDP settings and general MDPs with overparameterized neural network function approximation. We also showed empirically that the proposed framework could be both efficient and competitive for offline RL in some settings.

**Comparison with exact posterior sampling.** In both low-rank MDP and "neural" MDP settings that we considered, it is possible to use exact posterior sampling in PPS without resorting to LMC. We would like to make two remarks about this. First, we can show that PPS has the same-order bounds as LMC-PPS (but the analysis and algorithms for PPS are much simpler than those for LMC-PPS) for both settings. This shows that using approximate posterior sampling via LMC does not compromise the statistical guarantees of the exact posterior sampling in these settings. Second, computation-wise, in most cases exact posterior sampling is not even tractable, especially when the posterior does not admit any closed forms. LMC-PPS on the other hand can apply to any differentiable models, even beyond the low-rank MDPs and neural MDPs we considered. Even in the low-rank MDPs and the neural MDPs where the exact posterior sampling is computable, exact posterior sampling is computationally expensive in high dimensions and sometimes unstable. As a concrete comparison, in low-rank MDPs, the computational complexity of the exact posterior sampling is $\mathcal{O}(\min\{Kd^2, d^3 + Kd\})$ [11] (using the Sherman-Morrison formula to compute the inverse matrix) while that of Lin-LMC-PPS is $\mathcal{O}(\min\{d, K\} \cdot Kd)$. In high dimensions when $d \gg K$ (e.g., NTK setting), the latter is much smaller than the former.

**Theoretical gap between maximum likelihood methods and (approximate) PS for offline RL.** While maximum likelihood methods with pessimistic adjustments for offline RL can achieve a near-optimal bound (up to the natural $\mathcal{O}(\sqrt{H})$ gap due to the variance-agnostic nature of the algorithms) [ZWB21; XZS$^+$22], (approximate) posterior sampling currently suffers a gap of $\mathcal{O}(\sqrt{dH})$ (in the worst-case scenarios) in low-rank MDPs. Thus, we ask the follow-up question: Can (approximate) posterior sampling (e.g., via Langevin Monte Carlo) obtain optimal bounds in offline RL? We leave this important open question as a future work.

---

[11]In this specific case, it is possible to do exact posterior sampling with $\mathcal{O}(\min\{d, K\} \cdot Kd)$ computations using SVD and Woodbury lemma. However, it suffers from the numerical instability of SVD and the space complexity of $\Omega(dK)$ (to store the matrix for SVD) none of which LMC-PPS requires.

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

## CONTENTS

# Appendices

## APPENDIX A    PRELIMINARIES AND SUPPORTING LEMMAS

In this appendix, we present support lemmas that we frequently refer to during our proof process of the main theorems.

### A.1    NEURAL NETWORK APPROXIMATION

The following lemma quantifies the approximation error of representing an overparameterized neural network by its first-order Taylor approximation. It also bounds the 2-norm of the gradient of the network and with respect to the gradient at the initialization.

**Lemma A.1.** *Consider the neural network defined in Equation (1) with the symmetric initialization scheme considered in Section 2.2.2. Recall that $m$ is the network width, and $g(x; \theta) = \nabla_\theta f(x; \theta)$. Let $m = \Omega\left(d^{3/2} R^{-1} \ln^{3/2}(\sqrt{m}/R)\right)$ and $R = \mathcal{O}\left(m^{1/2} \ln^{-3} m\right)$. With probability at least $1 - e^{-\Omega(\ln^2 m)} \geq 1 - m^{-2}$ with respect to the random initialization, it holds for any $\theta, \theta' \in \mathcal{B}(\theta_0; R)$ and $x \in \mathbb{S}_{d-1}$ that*

$$\|g(x; \theta)\|_2 \leq C_g,$$

$$\|g(x; \theta) - g(x; \theta_0)\|_2 \leq \mathcal{O}\left(C_g R^{1/3} m^{-1/6} \sqrt{\ln m}\right),$$

$$|f(x; \theta) - f(x; \theta') - \langle g(x; \theta'), \theta - \theta' \rangle| \leq \mathcal{O}\left(C_g R^{4/3} m^{-1/6} \sqrt{\ln m}\right),$$

*where $C_g = \mathcal{O}(1)$ is a constant independent of $d$ and $m$. Moreover, without loss of generality, we assume $C_g \leq 1$.*

*Proof of Lemma A.1.* The proof directly follows from [YJW+20, Lemma C.2] or [CYLW19, Lemma F.1, F.2]. $\qquad\square$

The following lemma gives an exact characterization of using the network gradient at the initialization as a feature map when compared to using the NTK kernel.

**Lemma A.2** ([ADH+19]). *If $m = \Omega(\epsilon^{-4} \ln(1/\delta))$, then for any $x, x' \in \mathcal{X} \subset \mathbb{S}_{d-1}$, with probability at least $1 - \delta$,*

$$|\langle g(x; \theta_0), g(x'; \theta_0) \rangle - K_{ntk}(x, x')| \leq 2\epsilon.$$

### A.2    CONCENTRATION INEQUALITIES

The following lemma relates the in-expectation Bellman error to the empirical squared Bellman error, up to some estimation error that scales logarithmically with $K$. The logarithmic dependence on $K$ is crucial to avoid a vacuous bound if we pessimistically use the standard $\sqrt{K}$ estimation error. The key is to exploit the non-negativity of the squared Bellman error and use lemma A.6, which in turn uses the localization argument [BBM05] to obtain the fast rate in $K$.

**Lemma A.3** (Reduction to least-squares Bellman error). *For any $h \in [H]$, with probability at least $1 - 2\delta$, we have*

$$\mathbb{E}_\pi\left[\mathrm{err}_h(x_h)\right] \leq \frac{1}{\sqrt{K}} \sqrt{2 \sum_{k=1}^{K} C_\pi \mathrm{err}_h^2(x_h^k) + \frac{64 C_\pi H^2}{3} \ln \frac{\ln(4 K C_\pi H^2)}{\delta} + 2} + \sqrt{\frac{4H \ln(1/\delta)}{K}}.$$

*Proof of Lemma A.3.* Note that $\mathbb{E}_\pi\left[\mathrm{err}_h(s_h, a_h) \mid s_1^k\right]$ for $k \in [K]$ are i.i.d. random variables as $s_1^k \overset{\text{i.i.d.}}{\sim} d_1$ and that $|\mathbb{E}_\pi\left[\mathrm{err}_h(s_h, a_h) \mid s_1^k\right]| \leq 2H$. Thus, it follows from the tail inequality for sub-Gaussian variables $\mathbb{E}_\pi\left[\mathrm{err}_h(s_h, a_h) \mid s_1^k\right] \overset{\text{i.i.d.}}{\sim} \mathrm{subGaussian}(2H)$ that with probability at least $1 - \delta$,

$$
\begin{aligned}
\mathbb{E}_\pi\left[\mathrm{err}_h(s_h, a_h)\right] &\leq \frac{1}{K}\sum_{k=} \mathbb{E}_\pi\left[\mathrm{err}_h(s_h, a_h) \mid s_1 = s_1^k\right] + \sqrt{\frac{4H\ln(1/\delta)}{K}} \\
&\leq \frac{1}{K}\sqrt{K\sum_{k=1}^K \left\{\mathbb{E}_\pi\left[\mathrm{err}_h(s_h, a_h) \mid s_1 = s_1^k\right]\right\}^2} + \sqrt{\frac{4H\ln(1/\delta)}{K}} \\
&\leq \frac{1}{K}\sqrt{K\sum_{k=1}^K \mathbb{E}_\pi\left[\mathrm{err}_h^2(s_h, a_h) \mid s_1 = s_1^k\right]} + \sqrt{\frac{4H\ln(1/\delta)}{K}}, \quad (2)
\end{aligned}
$$

where the second inequality uses Cauchy-Schwartz inequality, and the third inequality uses Jensen's inequality. Denote the filtration $\mathcal{F}_{k-1} = \sigma\left(\{s_1^k\} \cup \{(s_h^k, a_h^k, r_h^k)\}_{h \in \times [H]}^{k \in [K]}\right)$. Let $Z_k = \frac{d_h^\pi(s_h^k, a_h^k)}{d_h^{\mu^k}(s_h^k, a_h^k)}\mathrm{err}_h^2(s_h^k, a_h^k)$. We have that $\mathbb{E}[Z_k|\mathcal{F}_{k-1}] = \mathbb{E}_\pi\left[\mathrm{err}_h^2(s_h, a_h) \mid s_1 = s_1^k\right]$, that $Z_k$ is $\mathcal{F}_k$-measurable (recall that the adaptive behaviour policy $\mu^k$ is $\mathcal{F}_k$-measurable) and that $Z_k \in [0, 4C_\pi H^2]$. Thus, it follows from Lemma A.6 that with probability at least $1 - \delta$, we have

$$
\begin{aligned}
\sum_{k=1}^K \mathbb{E}_\pi\left[\mathrm{err}_h^2(s_h, a_h) \mid s_1 = s_1^k\right] &= \sum_{k=1}^K \mathbb{E}_\pi\left[\mathrm{err}_h^2(s_h, a_h) \mid s_1 = s_1^k\right] \\
&\leq 2\sum_{k=1}^K Z_k + \frac{64C_\pi H^2}{3}\ln(\ln(4KC_\pi H^2)/\delta) + 2. \quad (3)
\end{aligned}
$$

Combing Equation (2) and Equation (3) via the union bound completes our proof. $\square$

The following lemma is the concentration of the self-normalizing process in RKHS.

**Lemma A.4** ([CG17, Theorem 1]). *Let $\mathcal{H}$ be an RKHS defined over $\mathcal{X} \subseteq \mathbb{R}^d$. Let $\{x_t\}_{t=1}^\infty$ be a discrete time stochastic process adapted to filtration $\{\mathcal{F}_t\}_{t=0}^\infty$. Let $\{Z_k\}_{k=1}^\infty$ be a real-valued stochastic process such that $Z_k \in \mathcal{F}_k$, and $Z_k$ is zero-mean and $\sigma$-sub Gaussian conditioned on $\mathcal{F}_{k-1}$. Let $E_k = (Z_1, \ldots, Z_{k-1})^T \in \mathbb{R}^{k-1}$ and $\mathcal{K}_k$ be the Gram matrix of $\mathcal{H}$ defined on $\{x_t\}_{t \leq k-1}$. For any $\rho > 0$ and $\delta \in (0, 1)$, with probability at least $1 - \delta$,*

$$
E_k^T\left[(\mathcal{K}_k + \rho I)^{-1} + I\right]^{-1} E_k \leq \sigma^2 \mathrm{logdet}\left[(1+\rho)I + \mathcal{K}_k\right] + 2\sigma^2\ln(1/\delta).
$$

The following lemma is a concentration of self-normalized processes in finite-dimensional Euclidean spaces.

**Lemma A.5** (Concentration of self-normalized processes [AYPS11]). *Let $\{\eta_t\}_{t=1}^\infty$ be a real-valued stochastic process with corresponding filtration $\{\mathcal{F}_t\}_{t=0}^\infty$ (i.e. $\eta_t$ is $\mathcal{F}_t$-measurable). Assume that $\eta_t|\mathcal{F}_{t-1}$ is zero-mean and $R$-subGaussian, i.e., $\mathbb{E}\left[\eta_t|\mathcal{F}_{t-1}\right] = 0$, and*

$$
\forall \lambda \in \mathbb{R}, \mathbb{E}\left[e^{\lambda \eta_t}|\mathcal{F}_{t-1}\right] \leq e^{\lambda^2 R^2/2}.
$$

*Let $\{x_t\}_{t=1}^\infty$ be an $\mathbb{R}^d$-valued stochastic process where $x_t$ is $\mathcal{F}_{t-1}$-measurable and $\|x_t\| \leq L$. Let $\Sigma_k = \lambda I_d + \sum_{t=1}^k x_t x_t^T$. Then for any $\delta > 0$, with probability at least $1 - \delta$, it holds for all $k > 0$ that*

$$
\left\|\sum_{t=1}^k x_t \eta_t\right\|_{\Sigma_k^{-1}}^2 \leq 2R^2\ln\left[\frac{\det(\Sigma_k)^{1/2}\det(\Sigma_0)^{-1/2}}{\delta}\right] \leq 2R^2\left[\frac{d}{2}\ln\frac{kL^2 + \lambda}{\lambda} + \ln\frac{1}{\delta}\right].
$$

The following lemma relates the summation of non-negative random variables to its in-expectation value, up to some estimation error that scales only logarithmically with the number of terms in the summation.

**Lemma A.6** ([NTYG+22]). *Let $\{X_k\}$ be any real-valued stochastic process adapted to the filtration $\{\mathcal{F}_k\}$, i.e. $X_k$ is $\mathcal{F}_k$-measurable. Suppose that for any $k$, $X_k \in [0, H]$ almost surely for some $H > 0$. For any $K > 0$, with probability at least $1 - \delta$, we have:*

$$\sum_{k=1}^{K} \mathbb{E}\left[X_k | \mathcal{F}_{k-1}\right] \leq 2 \sum_{k=1}^{K} X_k + \frac{16}{3} H \ln(\ln(KH)/\delta) + 2.$$

The following lemma is a concentration of the weighted norm of a high-dimension normal variable, weighted by its covariance matrix.

**Lemma A.7.** *Let $X \sim \mathcal{N}(0, a\Lambda^{-1})$ be a $d$-dimensional normal variable where $a$ is a scalar. There exists an absolute constant $c > 0$ such that for any $\delta > 0$, with probability at least $1 - \delta$,*

$$\|X\|_\Lambda \leq c\sqrt{da \ln(d/\delta)}.$$

*For $d = 1$, $c = \sqrt{2}$.*

## A.3 LINEAR ALGEBRA

The following lemma relates the difference between two inverse matrices to that of the original matrices.

**Lemma A.8.** *For any invertible matrices $A, B$,*

$$\|A^{-1} - B^{-1}\|_2 \leq \frac{\|A - B\|_2}{\lambda_{\min}(A)\lambda_{\min}(B)}.$$

*Proof of Lemma A.8.* We have:

$$
\begin{aligned}
\|A^{-1} - B^{-1}\|_2 &= \|(AB)^{-1}(AB)(A^{-1} - B^{-1})\|_2 = \|(AB)^{-1}(ABA^{-1} - A)\|_2 \\
&\leq \|(AB)^{-1}\|_2 \|ABA^{-1} - A\|_2 = \|(AB)^{-1}\|_2 \|ABA^{-1} - AAA^{-1}\|_2 \\
&= \|(AB)^{-1}\|_2 \|A(B - A)A^{-1}\|_2 = \|(AB)^{-1}\|_2 \|B - A\|_2 \\
&\leq \lambda_{\max}(A^{-1})\lambda_{\max}(B^{-1})\|_2 \|B - A\|_2.
\end{aligned}
$$

$\square$

# APPENDIX B   PROOF OF THEOREM 1

In this appendix, we provide a complete proof of Theorem 1.

## B.1 PREPARATION

We first set out stages for the our proof of Theorem 1, including notations and support lemmas. We use the following notations summarized in Table 1.

| | |
|---|---|
| $x_h^k$ | $(s_h^k, a_h^k)$ |
| $\Phi_h$ | $[\phi(s_h^k, a_h^k)]_{k \in [K]} \in \mathbb{R}^{d \times K}$ |
| $\Lambda_h$ | $\lambda I + \Phi_h \Phi_h^T$ |
| $A_h$ | $I - 2\eta \Lambda_h$ |
| $\kappa_h$ | $\frac{\lambda_{\max}(\Lambda_h)}{\lambda_{\min}(\Lambda_h)}$ |
| $y_h^k$ | $r_h^k + \tilde{V}_{h+1}(s_{h+1}^k)$ |
| $y_h$ | $[y_h^k]_{k \in [K]} \in \mathbb{R}^K$ |
| $\widehat{\theta}_h$ | $\Lambda_h^{-1}(\Phi_h y_h + \lambda \theta_0)$ |
| $\Sigma_{h,t}$ | $\tau(I - A_h^{2t})\Lambda_h^{-1}(I + A_h)^{-1}$ |
| $\text{err}_h(s,a)$ | $\mathbb{B}_h \widetilde{V}_{h+1}(x) - \widetilde{Q}_h(x)$ |

Table 1: Notations used in the setting of low-rank MDPs.

We first present a series of immediate lemmas that shall be used to construct the proof of Theorem 1. The following lemma asserts that the samples $\{\theta_h^i\}_{i \in [M]}$ from the Langevin dynamics are approximate samples from the data posterior.

**Lemma B.1.** *Conditioned on $\mathcal{D}$, we have*

$$\{\theta_h^i\}_{i \in [M]} \overset{i.i.d.}{\sim} \mathcal{N}(A_h^T \theta_0 + (I - A_h^T)\widehat{\theta}_h, \Sigma_{h,T}).$$

*In addition, if we set $\eta = \frac{1}{2\lambda_{\max}(\Lambda_h)}$, we have*

$$\tau\left(1 - (1 - \frac{2}{\kappa_h})^t\right)\Lambda_h^{-1} \preceq \Sigma_{h,t} \preceq \tau \Lambda_h^{-1}.$$

*Proof of Lemma B.1.* Consider any sequence $\{\theta_t\}_{t \geq 0}$ such that

$$\theta_t = \theta_{t-1} - \eta \nabla \hat{L}_h(\theta_{t-1}) + \sqrt{2\eta\tau}\epsilon_t, \text{ where } \epsilon_t \sim \mathcal{N}(0, 1), \forall t.$$

We have

$$\nabla \hat{L}_h(\theta) = 2(\Lambda_h \theta - \Phi_h y_h - \lambda \theta_0).$$

Thus, we have

$$
\begin{aligned}
\theta_t &= \theta_{t-1} - \eta \nabla \hat{L}_h(\theta_{t-1}) + \sqrt{2\eta\tau}\epsilon_t \\
&= A_h \theta_{t-1} + 2\eta(\Phi_h y_h + \lambda \theta_0) + \sqrt{2\eta\tau}\epsilon_t \\
&= A_h^t \theta_0 + 2\eta \sum_{l=0}^{t-1} A_h^l (\Phi_h y_h + \lambda \theta_0) + \sqrt{\eta\tau} \sum_{l=0}^{t-1} A_h^l \epsilon_{t-l} \\
&= A_h^t \theta_0 + 2\eta(I - A_h^t)(I + A_h)^{-1}(I - A_h)^{-1} (\Phi_h y_h + \lambda \theta_0) + \sqrt{2\eta\tau} \sum_{l=0}^{t-1} A_h^l \epsilon_{t-l} \\
&= A_h^t \theta_0 + (I - A_h^t)\widehat{\theta}_h + \sqrt{2\eta\tau} \sum_{l=0}^{t-1} A_h^l \epsilon_{t-l}.
\end{aligned}
$$

Since $\{\epsilon_t\}_{t \geq 0}$ are mutually independent Gaussian noises, we have

$$\theta_t \sim \mathcal{N}(A_h^t \theta_0 + (I - A_h^t)\widehat{\theta}_h, \Sigma_{h,t}).$$

In addition, if we set $\eta = \frac{1}{2\lambda_{\max}(\Lambda_h)}$, we have

$$\tau\left(1 - (1 - \frac{2}{\kappa_h})^t\right)\Lambda_h^{-1} \preceq \Sigma_{h,t} \preceq \tau\Lambda_h^{-1}.$$

$\square$

The following lemma gives upper bounds on the 2-norm of the regularized least-squares minimizer $\widehat{\theta}_h$ and the parameters learned by the Langevin dynamics $\{\theta_h^i\}_{i \in [M]}$.

**Lemma B.2.** *We have*

$$\|\widehat{\theta}_h\|_2 \leq \frac{(H - h + 1)(1 + \iota)K}{\sqrt{\lambda}}.$$

*If we choose $\theta_0 = 0$ and $\eta\lambda < 1/2$, for any $i \in [M]$, with probability at least $1 - \delta$, we have*

$$\|\theta_h^i\|_2 \leq (1 - (1 - 2\eta\lambda)^T)\frac{(H - h + 1)(1 + \iota)K}{\sqrt{\lambda}} + c_1\sqrt{\frac{d}{\lambda}(1 + \ln(1/\delta))}$$

*for some absolute constant $c_1 > 0$.*

The following lemma bounds the uncertainty of the regularized least-squares minimizer $\widehat{\theta}_h$ in estimating the Bellman operator.

**Lemma B.3.** *If we choose $\theta_0 = 0$ and $\eta\lambda < 1/2$, then with probability at least $1 - \delta$, it holds uniformly for all $(h, s, a) \in [H] \times \mathcal{S} \times \mathcal{A}$ that*

$$|\mathbb{B}_h \widetilde{V}_{h+1}(s, a) - \langle \phi_h(s, a), \widehat{\theta}_h \rangle| \leq \gamma \|\phi_h(s, a)\|_{\Lambda_h^{-1}}$$

*where*

$$\gamma := 2H\sqrt{\lambda d} + \frac{2H}{\sqrt{\lambda}} + H\sqrt{d}\sqrt{\ln(1 + K/\lambda) + 2\ln(1 + 2RK) + 2\ln(2/\delta)},$$

$$R = (1 - (1 - 2\eta\lambda)^T)\frac{(H - h + 1)(1 + \iota)K}{\sqrt{\lambda}} + \sqrt{\frac{d}{\lambda}(1 + \ln(2M/\delta))}.$$

*Proof of Lemma B.3.* Due to the low-rank MDP assumption in Definition 1, there exists some $\widetilde{\theta}_h \in \mathbb{R}^d$ such that

$$\mathbb{B}_h \widetilde{V}_{h+1}(s, a) = \langle \phi_h(s, a), \widetilde{\theta}_h \rangle, \text{ and } \|\widetilde{\theta}_h\|_2 \leq \sqrt{d}(H - h + 2) \leq 2H\sqrt{d}.$$

Thus, we have

$$\mathbb{B}_h \widetilde{V}_{h+1}(s, a) - \langle \phi_h(s, a), \widehat{\theta}_h \rangle = \phi_h(s, a)^T \widetilde{\theta}_h - \phi_h(s, a)^T \Lambda_h^{-1} \sum_{k=1}^K \phi_h(x_h^k)(r_h^k + \widetilde{V}_{h+1}(s_{h+1}^k))$$

$$= \phi_h(s, a)^T \widetilde{\theta}_h - \phi_h(s, a)^T \Lambda_h^{-1} \sum_{k=1}^K \phi_h(x_h^k)\phi_h(x_h^k)^T \widetilde{\theta}_h$$

$$+ \phi_h(s, a)^T \Lambda_h^{-1} \sum_{k=1}^K \phi_h(x_h^k)(r_h^k + \widetilde{V}_{h+1}(s_{h+1}^k) - \mathbb{B}_h \widetilde{V}_{h+1}(x_h^k))$$

$$= \lambda \phi_h(s, a)^T \Lambda_h^{-1} \widetilde{\theta}_h + \phi_h(s, a)^T \Lambda_h^{-1} \sum_{k=1}^K \phi_h(x_h^k)(r_h^k + \widetilde{V}_{h+1}(s_{h+1}^k) - \mathbb{B}_h \widetilde{V}_{h+1}(x_h^k))$$

$$\leq \lambda \|\phi_h(s,a)\|_{\Lambda_h^{-1}} \|\widetilde{\theta}_h\|_{\Lambda_h^{-1}} + \|\phi_h(s,a)\|_{\Lambda_h^{-1}} \|\sum_{k=1}^K \phi_h(x_h^k)(r_h^k + \widetilde{V}_{h+1}(s_{h+1}^k) - \mathbb{B}_h \widetilde{V}_{h+1}(x_h^k))\|_{\Lambda_h^{-1}}$$

We have

$$\|\widetilde{\theta}_h\|_{\Lambda_h^{-1}} \leq \sqrt{\|\Lambda_h^{-1}\|_2} \|\widetilde{\theta}_h\|_2 \leq \sqrt{\frac{d}{\lambda}}(H - h + 2).$$

Now we bound the term $L(\widetilde{V}_{h+1})$ where

$$L(V) := \|\sum_{k=1}^K \phi_h(x_h^k)(r_h^k + V(s_{h+1}^k) - \mathbb{B}_h V(x_h^k))\|_{\Lambda_h^{-1}},$$

for any $V \in \{\mathcal{S} \to \mathbb{R}\}$. Now fix any $\delta > 0$. Consider the event

$$E = \left\{ \|\theta_h^i\|_2 \lesssim (1 - (1 - 2\eta\lambda)^T)\frac{(H - h + 1)K}{\sqrt{\lambda}} + \sqrt{\frac{d}{\lambda}}(1 + \ln(2M/\delta)), \forall i \in [M] \right\}.$$

By Lemma B.2 and the union bound, we have $\Pr(E) \geq 1 - \delta/2$. In the rest of our proof, we consider everything under event $E$. Let us define

$$\mathcal{V} := \left\{ \min\{\max_{a \in \mathcal{A}}\langle\phi_{h+1}(\cdot, a), \theta\rangle, H - h\}^+ : \|\theta\|_2 \lesssim (1 - (1 - 2\eta\lambda)^T)\frac{(H - h + 1)(1 + \iota)K}{\sqrt{\lambda}} \right.$$
$$\left. + \sqrt{\frac{d}{\lambda}}(1 + \ln(2M/\delta)) \right\}.$$

It is easy to verify that for any $V', V \in \mathcal{V}$, we have

$$|L(V') - L(V)| \leq 2\frac{K}{\sqrt{\lambda}}(H - h + 1)(1 + \iota)\|V' - V\|_\infty.$$

For any fixed $V \in \mathcal{V}$, $r_h^k + V(s_{h+1}^k) - \mathbb{B}_h V(x_h^k)$ is zero-mean and $(H - h + 1)(1 + \iota)$-subGaussian conditioned on the offline data prior to episode $k$. Thus, by Lemma A.5, with probability at least $1 - \delta$, we have

$$L(V) \leq (H - h + 1)(1 + \iota)\sqrt{d\ln(1 + K/\lambda) + 2\ln(1/\delta)}.$$

Since $\widetilde{V}_{h+1} \in \mathcal{V}$ (under event $E$), there exists $V'$ in the $\epsilon$-cover of $\mathcal{V}$ such that $\|\widetilde{V}_{h+1} - V'\|_\infty \leq \epsilon$. By the union bound, with probability $1 - \delta$, we have

$$L(\widetilde{V}_{h+1}) \leq L(V') + |L(\widetilde{V}_{h+1}) - L(V')|$$
$$= \sup_{V' \in \epsilon\text{-cover of } \mathcal{V}} L(V') + \frac{2K}{\sqrt{\lambda}}(H - h + 1)(1 + \iota)\epsilon$$
$$\leq (H - h + 1)(1 + \iota)\sqrt{d\ln(1 + K/\lambda) + 2\ln(N(\mathcal{V}, \epsilon)/\delta)} + \frac{2K}{\sqrt{\lambda}}(H - h + 1)(1 + \iota)\epsilon,$$

where $N(\mathcal{V}, \epsilon)$ is the $\epsilon$-covering number of $\mathcal{V}$ with respect to norm $\|\cdot\|_\infty$. We have

$$N(\mathcal{V}, \epsilon) \leq (1 + 2R/\epsilon)^d$$

where $R = (1 - (1 - 2\eta\lambda)^T)\frac{(H-h+1)(1+\iota)K}{\sqrt{\lambda}} + \sqrt{\frac{d}{\lambda}}(1 + \ln(2M/\delta))$.

Set $\epsilon = 1/K$ and combining all pieces using the union bound yields the claim.

$\square$

The following lemma asserts (approximate) point-wise pessimism, i.e., the Bellman error $\text{err}_h(s, a)$ is nearly non-negative for all state-action pair for all timestep $h \in [H]$.

**Lemma B.4** (Pessimism). *If we set $M = \ln \frac{HS}{\delta} / \ln \frac{1}{1 - \Phi(-1)}$ and $\sqrt{(1 - (1 - \frac{1}{2\kappa_h})^T)\tau} \geq \gamma$ where $\gamma$ is defined in Lemma B.3, then with probability at least $1 - \delta$, it holds uniformly for all $(h, s) \in [H] \times \mathcal{S}$ that*

$$\mathrm{err}_h(s, a) \geq -(1 - 2\eta\lambda)^T HK / \sqrt{\lambda}.$$

*Proof of Lemma B.4.* With probability at least $1 - \delta$, for any $s \in \mathcal{S}$, we have

$$\widetilde{Q}_h(s, \widetilde{\pi}_h(s)) = \min\{\min_{i \in [M]} \langle \phi_h(s, \widetilde{\pi}_h(s)), \theta_h^i \rangle, (H - h + 1)(1 + \iota)\}^+$$
$$\leq \langle \phi_h(s, \widetilde{\pi}_h(s)), (I - A_h^T)\widehat{\theta}_h \rangle - \|\phi_h(s, \widetilde{\pi}_h(s))\|_{\Sigma_h},$$

Thus, by the union bound, with probability at least $1 - \delta$, it holds uniformly for all $h, s$ that

$$\mathrm{err}_h(s, \widetilde{\pi}_h(s)) = \mathbb{B}_h\widetilde{V}_{h+1}(s, \widetilde{\pi}_h(s)) - \widetilde{Q}_h(s, \widetilde{\pi}_h(s))$$
$$\geq \mathbb{B}_h\widetilde{V}_{h+1}(s, \widetilde{\pi}_h(s)) - \langle \phi_h(s, \widetilde{\pi}_h(s)), (I - A_h^T)\widehat{\theta}_h \rangle + \|\phi_h(s, \widetilde{\pi}_h(s))\|_{\Sigma_h}$$
$$\geq \langle \phi_h(s, \widetilde{\pi}_h(s)), \widehat{\theta}_h \rangle - \gamma\|\phi_h(s, \widetilde{\pi}_h(s))\|_{\Lambda_h^{-1}} - \langle \phi_h(s, \widetilde{\pi}_h(s)), (I - A_h^T)\widehat{\theta}_h \rangle + \|\phi_h(s, \widetilde{\pi}_h(s))\|_{\Sigma_h}$$
$$\geq \langle \phi_h(s, \widetilde{\pi}_h(s)), A_h^T\widehat{\theta}_h \rangle + \left( \sqrt{(1 - (1 - \frac{1}{2\kappa_h})^T)\tau} - \gamma \right) \|\phi_h(s, \widetilde{\pi}_h(s))\|_{\Lambda_h^{-1}}$$
$$\geq -(1 - 2\eta\lambda)^T HK / \sqrt{\lambda} + \left( \sqrt{(1 - (1 - \frac{1}{2\kappa_h})^T)\tau} - \gamma \right) \|\phi_h(s, \widetilde{\pi}_h(s))\|_{\Lambda_h^{-1}}$$

where the second inequality uses Lemma B.3, the third inequality uses Lemma B.1, and the fourth inequality uses Lemma B.2. □

## B.2 PROOF OF THEOREM 1

We are now ready to prove Theorem 1.

*Proof of Theorem 1.* For notational convenience, we define the following quantities: where

$$\gamma(\delta) := 2H\sqrt{\lambda d} + \frac{2H}{\sqrt{\lambda}} + H\sqrt{d}\sqrt{\ln(1 + K/\lambda) + 2\ln(1 + 2R(\delta)K) + 2\ln(2/\delta)},$$
$$R(\delta) := (1 - (1 - 2\eta\lambda)^T)\frac{(H - h + 1)(1 + \iota)K}{\sqrt{\lambda}} + \sqrt{\frac{d}{\lambda}(1 + \ln(2M/\delta))},$$
$$M(\delta) := \ln \frac{H|\mathcal{S}|}{\delta} / \ln \frac{1}{1 - \Phi(-1)}.$$

Suppose that we set the parameters as follows: $\sqrt{(1 - (1 - \frac{1}{2\kappa_h})^T)\tau} \geq \gamma(\delta/4), \eta\lambda < 1/2, M = M(\delta/4), \iota \geq (1 - 2\eta\lambda)^T HK / \sqrt{\lambda}$.

We start with the error decomposition:

$$\mathrm{SubOpt}_\pi(\widetilde{\pi}) \leq \sum_{h=1}^H \mathbb{E}_\pi \left[ \mathrm{err}_h(s_h, a_h) \right] - \sum_{h=1}^H \mathbb{E}_{\widetilde{\pi}} \left[ \mathrm{err}_h(s_h, a_h) \right], \tag{4}$$

where $\mathrm{err}_h(x) := \mathbb{B}_h\widetilde{V}_{h+1}(x) - \widetilde{Q}_h(x)$. Consider the following event $E = E_1 \cap E_2 \cap E_3 \cap E_4$, where

$$E_1 = \left\{ |\mathbb{B}_h\widetilde{V}_{h+1}(s, a) - \langle \phi_h(s, a), \widehat{\theta}_h \rangle| \leq \gamma\|\phi_h(s, a)\|_{\Lambda_h^{-1}}, \forall(s, a, h) \in \mathcal{S} \times \mathcal{A} \times [H] \right\},$$
$$E_2 = \left\{ \mathrm{err}_h(s, a) \geq -(1 - 2\eta\lambda)^T HK\lambda^{-1/2}, \forall(h, s, a) \in [H] \times \mathcal{S} \times \mathcal{A} \right\},$$

$$E_3 = \left\{ \mathbb{E}_\pi \left[\text{err}_h(x_h)\right] \leq \frac{1}{\sqrt{K}} \sqrt{2 \sum_{k=1}^{K} C_\pi \text{err}_h^2(x_h^k) + \frac{64 C_\pi H^2}{3} \ln \frac{\ln(4 K C_\pi H^2)}{\delta} + 2} \right.$$

$$\left. + \sqrt{\frac{4H \ln(1/\delta)}{K}}, \forall h \in [H] \right\},$$

$$E_4 = \left\{ \langle \phi_h(x_h^k), (I - A_h^T)\widehat{\theta}_h - \widetilde{\theta}_h^i \rangle \leq \|\phi_h(x_h^k)\|_{\Sigma_h} \sqrt{2 \ln(KHM/\delta)} \right.$$

$$\left. : \forall (k, h, i) \in [K] \times [H] \times [M] \right\}.$$

Under $E_1 \cap E_3$, for any $(k, h) \in [K] \times [H]$, we have

$$\text{err}_h(x_h^k) \leq \mathbb{B}_h \widetilde{V}_{h+1}(x_h^k) - \langle \phi_h(x_h^k), \widehat{\theta}_h \rangle + \max_{i \in [M]} \langle \phi_h(x_h^k), (I - A_h^T)\widehat{\theta}_h - \widetilde{\theta}_h^i \rangle + \langle \phi_h(x_h^k), A_h^T \widehat{\theta}_h \rangle$$

$$\leq (\gamma + \tau \sqrt{2 \ln(KHM/\delta)}) \|\phi_h(x_h^k)\|_{\Lambda_h^{-1}} + (1 - 2\eta\lambda)^T \frac{(H - h + 1)(1 + \iota)K}{\sqrt{\lambda}}$$

By Lemma B.1 and Lemma A.7, we have $\Pr(E_4|\mathcal{D}) \geq 1 - \delta$. Thus, we have $\Pr(E_4) = \sum_\mathcal{D} \Pr(E_4|\mathcal{D}) \Pr(\mathcal{D}) \geq 1 - \delta$. By Lemma B.3, Lemma B.4, and Lemma A.3, we have $\Pr(E_1) \geq 1 - \delta, \Pr(E_2) \geq 1 - \delta$, and $\Pr(E_3) \geq 1 - \delta$, respectively. Thus, by the union bound, we have

$$\Pr(E) \geq 1 - 4\delta.$$

Under event $E_2$, we have

$$\widetilde{Q}_h(s, a) = \mathbb{B}_h \widetilde{V}_{h+1}(s, a) - \text{err}_h(s, a)$$
$$\leq 1 + (H - h)(1 + \iota) + (1 - 2\eta\lambda)^T HK/\sqrt{\lambda}$$
$$\leq (H - h + 1)(1 + \iota),$$

where the first inequality uses Lemma B.4 and the second inequality uses the choice $\iota \geq (1 - 2\eta\lambda)^T HK/\sqrt{\lambda}$. Thus, under event $E_2$, we have

$$\widetilde{Q}_h(\cdot, \cdot) = \min\{\min_{i \in [M]} \langle \phi_h(\cdot, \cdot), \widetilde{\theta}_h^i \rangle, (H - h + 1)(1 + \iota)\}^+ = \max\{\min_{i \in [M]} \langle \phi_h(\cdot, \cdot), \widetilde{\theta}_h^i \rangle, 0\}.$$

Hence, under event $E_1 \cap E_2 \cap E_4$, we have

$$\text{err}_h(x_h^k) = \mathbb{B}_h \widetilde{V}_{h+1}(x_h^k) - \widetilde{Q}_h(x_h^k)$$

$$= \mathbb{B}_h \widetilde{V}_{h+1}(x_h^k) - \max\{\min_{i \in [M]} \langle \phi_h(x_h^k), \widetilde{\theta}_h^i \rangle, 0\}$$

$$\leq \mathbb{B}_h \widetilde{V}_{h+1}(x_h^k) - \min_{i \in [M]} \langle \phi(x_h^k), \widetilde{\theta}_h^i \rangle$$

$$= \mathbb{B}_h \widetilde{V}_{h+1}(x_h^k) - \langle \phi(x_h^k), \widehat{\theta}_h \rangle - \min_{i \in [M]} \langle \phi(x_h^k), \widetilde{\theta}_h^i - A_h^T \theta_0 - (I - A_h^T)\widehat{\theta}_h \rangle + A_h^T(\widehat{\theta}_h - \theta_0) \rangle$$

$$\leq (\gamma + \sqrt{2\tau \ln(KHM/\delta)}) \|\phi_h(x_h^k)\|_{\Lambda_h^{-1}} + (1 - 2\eta\lambda)^T \frac{(H - h + 1)(1 + \iota)K}{\sqrt{\lambda}},$$

where the last inequality uses $E_1$, $E_2$ and $E_4$, and Lemma B.2. Combing with $E_3$, we have that under $E$,

$$\mathbb{E}_\pi[\text{err}_h(x_h)] \lesssim \sqrt{\frac{C_\pi}{K}}(\gamma + \sqrt{2\tau \ln(KHM/\delta)}) \sqrt{\sum_{k=1}^{K} \|\phi_h(x_h^k)\|_{\Lambda_h^{-1}}^2}$$

$$+ \sqrt{C_\pi}(1 - 2\eta\lambda)^T \frac{(H - h + 1)(1 + \iota)K}{\sqrt{\lambda}} + H\sqrt{\frac{C_\pi}{K} \ln \frac{\ln K C_\pi H^2}{\delta}}$$

$$+ \sqrt{\frac{H \ln(1/\delta)}{K}}.$$

Finally, note that

$$
\begin{aligned}
\sum_{k=1}^{K} \|\phi_h(x_h^k)\|_{\Lambda_h^{-1}}^2 &= \sum_{k=1}^{K} \phi_h(x_h^k)^T \Lambda_h^{-1} \phi_h(x_h^k) \\
&= \sum_{k=1}^{K} \mathrm{tr}\left(\phi_h(x_h^k)^T \Lambda_h^{-1} \phi_h(x_h^k)\right) \\
&= \sum_{k=1}^{K} \mathrm{tr}\left(\phi_h(x_h^k)\phi_h(x_h^k)^T \Lambda_h^{-1}\right) \\
&= \mathrm{tr}\left(\sum_{k=1}^{K} \phi_h(x_h^k)\phi_h(x_h^k)^T \Lambda_h^{-1}\right) \\
&= \mathrm{tr}((\Lambda_h - \lambda I)\Lambda_h^{-1}) \\
&= \mathrm{tr}(I - \lambda \Lambda_h^{-1}) \\
&= d - \sum_{i=1}^{d} \frac{\lambda}{\lambda_i(\Lambda_h)},
\end{aligned}
$$

where recall that $\{\lambda_i(\Lambda_h)\}_{i\in[d]}$ is the eigenvalues of $\Lambda_h$.

□

## APPENDIX C   PROOF OF THEOREM 2

We now construct the full proof of Theorem 2. For convenience, in this appendix, we shall write $E\{\text{Lemma x}\}$ to denote the event in which the statement of Lemma x holds, and $E\{\text{Equation (x)}\}$ to denote the event in which Equation (x) holds.

### C.1   PREPARATION

GRADIENT DESCENT OF A NEW AUXILIARY LINEAR MODEL AT THE INITIAL PARAMETERS

We consider the linear objective function

$$
\bar{\mathcal{L}}_h^{lin}(\theta) := \sum_{k\in\mathcal{I}_h} \left(\langle g(x_h^k;\theta_0),\theta\rangle - r_h^k - \widetilde{V}_{h+1}(s_{h+1}^k)\right)^2 + \lambda\|\theta - \theta_0\|_2^2.
$$

Let $\bar{\theta}_h^{lin} \leftarrow \mathrm{GD}(\bar{\mathcal{L}}_h^{lin}, \theta_0, \eta, T_1)$ whose update is unrolled as

$$
\bar{\theta}_{h,t}^{lin} \leftarrow \bar{\theta}_{h,t-1}^{lin} - \eta_1 \nabla \mathcal{L}_h(\bar{\theta}_{h,t-1}), \forall t \in [T_1], \tag{5}
$$

where $\bar{\theta}_{h,0}^{lin} = \theta_0$ and $\bar{\theta}_h^{lin} = \bar{\theta}_{h,T_2}^{lin}$. We also define the following notations:

$$
\Lambda_h := \lambda I + \sum_{k\in\mathcal{I}_h} g(x_h^k;\theta_h)g(x_h^k;\theta_h)^T \in \mathbb{R}^{dm\times dm},
$$

$$
\bar{\Lambda}_h := \lambda I + \sum_{k\in\mathcal{I}_h} g(x_h^k;\theta_0)g(x_h^k;\theta_0)^T \in \mathbb{R}^{dm\times dm},
$$

$$
A_h := I - 2\eta_2 \Lambda_h \in \mathbb{R}^{dm\times dm},
$$

$$
G_h(\theta) := (g(x_h^k;\theta))_{k\in\mathcal{I}_h} \in \mathbb{R}^{dm\times K'},
$$

$$
y_h := (r_h^k + \widetilde{V}_{h+1}(s_{h+1}^k))_{k\in\mathcal{I}_h} \in \mathbb{R}^{K'}.
$$

ADDITIONAL NOTATIONS

Let $\hat{\theta}_h^{lin}$ and $\hat{\bar{\theta}}_h^{lin}$ be the minimizers of $\mathcal{L}_h^{lin}(\theta)$ and $\bar{\mathcal{L}}_h^{lin}(\theta)$, respectively, i.e.,

$$
\hat{\theta}_h^{lin} = \Lambda_h^{-1}\left(G(\hat{\theta}_h)y_h + \lambda\theta_0\right) \text{ and } \hat{\bar{\theta}}_h^{lin} = \bar{\Lambda}_h^{-1}\left(G(\theta_0)y_h + \lambda\theta_0\right). \tag{6}
$$

For convenience, we summarize all notations, old and new, into Table 2.

C.2   TRAINING DYNAMICS IN THE NEURAL TANGENT KERNEL (NTK) REGIME

In this part, we establish the optimization guarantee of training a neural value function with GD under the NTK regime. Our analysis builds up on the recent advances in understanding the training dynamics of (S)GD of overparameterized neural networks through the prism of Neural Tangent Kernel (NTK) [JGH18] and its application to the analysis of neural UCB [ZLG20]. Though we partly borrowed the proof flow and some techniques from [ZLG20] for our own proof, the key distinction of our analysis with that of [ZLG20] is that, due to the different nature of the offline learning vs the online learning, we need to construct a network as a value predictor in *all* state-action pairs, while [ZLG20] only work on the empirical state-action pairs that the online learner acquires. Thus, we rely on the first-order Taylor approximation to the neural network for all state-action pairs, a result that is presented in [YJW$^+$20, Lemma C.2] or [CYLW19, Lemma F.1, F.2].

Another key distinction in our setting is that we analyze the general MDP setting, instead of the contextual bandit setting. The MDP setting gives rise to a distinct challenge of dealing with a data-dependent regression target, wherein each timestep $h \in [H]$, we perform regression on mapping each $(s_h^k, a_h^k)$ to $r_h^k + \widetilde{V}_{h+1}(s_{h+1}^k)$, where $\widetilde{V}_{h+1}$ is a value estimate from step $h+1$. If we use up all offline data to obtain $\widetilde{V}_{h+1}$, then $\widetilde{V}_{h+1}$ depends on $(s_h^k, a_h^k)$. Thus,

$$\mathbb{E}\left[\widetilde{V}_{h+1}(s_{h+1}^k)\right] \neq \mathbb{E}\left[(\mathbb{B}_h \widetilde{V}_{h+1})(s_h^k, a_h^k)\right].$$

To handle this statistical dependence issue, we typically use a uniform convergence argument where we involve uniformly over all $V_{h+1}$ in the function class where $\widetilde{V}_{h+1}$ belongs to. Unfortunately, the uniform convergence argument does not work for the overparameterized setting. Concretely, the uniform convergence argument scales the sub-optimality bound with the effective size of the function class – which scales polynomially with $K$ and thus become vacuous. We completely avoid this issue by splitting the offline data into $H$ different folds, where each timestep $h \in [H]$ uses a different data fold for estimation. This data splitting technique follows from [NTA23].

In what follows, we present a series of immediate results that culminate into our proof construction for Theorem 2. We start with the following lemma which indicates that the network weights after gradient updates tend to stay close to the initialization, under certain conditions.

**Lemma C.1.** *We introduce an additional parameter $R = \mathcal{O}\left(m^{1/2} \ln^{-3} m\right)$ and suppose that*

$$m = \Omega\left(d^{3/2} R^{-1} \ln^{3/2}(\sqrt{m}/R)\right),$$

$$(\eta_1 \lambda)^{-1} R^{8/3} m^{-1/3} \ln m \lesssim 1,$$

$$\eta_1(K' C_g^2 + \lambda/2) \leq 1/2,$$

$$R \gtrsim \lambda^{-1}(K'(H-h+1)^2(1+\iota)^2 + 1)\sqrt{K'} R^{1/3} m^{-1/6}\sqrt{m}$$

$$+ \lambda^{-1} K' C_g R^{4/3} m^{-1/6}\sqrt{\ln m} + \lambda^{-1}\sqrt{K'} H(1+\iota),$$

*where $C_g$ is an absolute constant in Lemma A.1. Then, with probability at least $1 - m^{-2}$, for any t, we have*

$$\theta_{h,t} \in \mathcal{B}(\theta_0; R) := \{\theta \in \mathbb{R}^{md} : \|\theta - \theta_0\|_2 \leq R\}, \text{ and}$$

$$\|\theta_{h,t} - \bar{\theta}_{h,t}^{lin}\|_2 \lesssim \lambda^{-1}(K'(H-h+1)^2(1+\iota)^2 + 1)\sqrt{K'} R^{1/3} m^{-1/6}\sqrt{m} + \lambda^{-1} K' C_g R^{4/3} m^{-1/6}\sqrt{\ln m}.$$

*Proof of Lemma C.1.* Consider a fixed $h \in [H]$. For simplicity, we define

$$\Delta_t := \theta_{h,t} - \bar{\theta}_{h,t}^{lin}$$

$$G_t := (g(x_h^k; \theta_{h,t}))_{k \in \mathcal{I}_h} \in \mathbb{R}^{md \times K'}$$

$$H_t := G_t G_t^T \in \mathbb{R}^{md \times md}$$

$$f_t := \left(f(x_h^k; \theta_{h,t})\right)_{k \in \mathcal{I}_h} \in \mathbb{R}^{K'},$$

| | |
|---|---|
| $x_h^k$ | $(s_h^k, a_h^k)$ |
| $y_h^k$ | $r_h^k + \widetilde{V}_{h+1}(s_{h+1}^k)$ |
| $y_h$ | $[y_h^k]_{k \in [K]} \in \mathbb{R}^K$ |
| $\mathrm{err}_h(s, a)$ | $\mathbb{B}_h \widetilde{V}_{h+1}(x) - \widetilde{Q}_h(x)$ |
| $f(x; \theta)$ | $\frac{1}{\sqrt{m}} \sum_{i=1}^m b_i \sigma(w_i^T x)$ defined in Equation (1) |
| $g(x; \theta)$ | $\nabla_\theta f(x; \theta)$ |
| $\Lambda_h$ | $\lambda I + \sum_{k \in \mathcal{I}_h} g(x_h^k; \theta_h) g(x_h^k; \theta_h)^T \in \mathbb{R}^{dm \times dm}$ |
| $\bar{\Lambda}_h$ | $\lambda I + \sum_{k \in \mathcal{I}_h} g(x_h^k; \theta_0) g(x_h^k; \theta_0)^T \in \mathbb{R}^{dm \times dm}$ |
| $A_h$ | $I - 2\eta_2 \Lambda_h \in \mathbb{R}^{dm \times dm}$ |
| $G_h(\theta)$ | $(g(x_h^k; \theta))_{k \in \mathcal{I}_h} \in \mathbb{R}^{dm \times K'}$ |
| $\mathcal{L}_h(\theta)$ | $\sum_{k \in \mathcal{I}_h} (f(x_h^k; \theta) - r_h^k - \widetilde{V}_{h+1}(s_{h+1}^k))^2 + \lambda \|\theta - \theta_0\|_2^2$ |
| $\theta_{h,t}$ | $\theta_{h,t-1} - \eta_1 \nabla \mathcal{L}_h(\theta_{h,t-1})$ |
| $\theta_h$ | $\theta_{h,T_1}$ |
| $\mathcal{L}_h^{lin}(\theta)$ | $\sum_{k \in \mathcal{I}_h} (\theta^T g(x_h^k; \theta_h) - r_h^k - \widetilde{V}_{h+1}(s_{h+1}^k))^2 + \lambda \|\theta - \theta_0\|_2^2$ |
| $\theta_{h,t}^{lin}$ | $\theta_{h,t-1}^{lin} - \eta_2 \nabla \mathcal{L}_h^{lin}(\theta_{h,t-1}^{lin})$ |
| $\theta_h^{lin}$ | $\theta_{h,T_2}^{lin}$ |
| $\theta_{h,t}^{lin,i}$ | $\theta_{h,t-1}^{lin,i} - \eta_2 \nabla \mathcal{L}_h^{lin}(\theta_{h,t-1}^{lin,i}) + \sqrt{2\eta\tau}\epsilon_t$ |
| $\theta_h^{lin,i}$ | $\theta_{h,T_2}^{lin,i}$ |
| $\bar{\mathcal{L}}_h^{lin}(\theta)$ | $\sum_{k \in \mathcal{I}_h} \left( \langle g(x_h^k; \theta_0), \theta \rangle - r_h^k - \widetilde{V}_{h+1}(s_{h+1}^k) \right)^2 + \lambda \|\theta - \theta_0\|_2^2$ |
| $\bar{\theta}_{h,t}^{lin}$ | $\bar{\theta}_{h,t-1}^{lin} - \eta_1 \nabla \mathcal{L}_h(\bar{\theta}_{h,t-1})$ |
| $\bar{\theta}_h^{lin}$ | $\bar{\theta}_{h,T_2}^{lin}$ |

Table 2: Notations used in the setting of general MDPs with neural function approximation.

where we drop the dependence on $h$ for notational simplicity. For any $\theta \in \mathbb{R}^{md}$, we also write

$$G_\theta := (g(x_h^k; \theta))_{k \in \mathcal{I}_h} \in \mathbb{R}^{md \times K'},$$
$$f_\theta := \left( f(x_h^k; \theta) \right)_{k \in \mathcal{I}_h} \in \mathbb{R}^{K'}.$$

We re-write the GD update in Line 4 of Algorithm 3 and that in Equation (5), respectively, as

$$\theta_{h,t} = \theta_{h,t-1} - 2\eta_1(G_{t-1}(f_{t-1} - y_h) + \lambda(\theta_{h,t-1} - \theta_0)),$$

$$\bar{\theta}_{h,t}^{lin} = \bar{\theta}_{h,t-1}^{lin} - 2\eta_1(G_0(G_0^T(\bar{\theta}_{h,t-1}^{lin} - \theta_0) - y_h) + \lambda(\bar{\theta}_{h,t-1}^{lin} - \theta_0)).$$

We will prove the lemma by induction with $t$. The statement of the lemma obviously holds at $t = 0$, since $\theta_{h,0} = \bar{\theta}_{h,0}^{lin} = \theta_0$. Assume the lemma statement holds up to some $t - 1$. We will prove that it also holds for $t$. Indeed, let us consider the rest of this proof under event $E\{\text{Lemma A.1}\}$. We have

$$\|\Delta_t\|_2$$
$$= \left\| (1 - \eta_1\lambda)\Delta_{t-1} - \eta_1\left[G_0(f_{t-1} - G_0^T(\theta_{h,t-1} - \theta_0)) + G_0G_0^T(\theta_{h,t-1} - \bar{\theta}_{h,t-1}^{lin}) + (f_{t-1} - y_h)(G_{t-1} - G_0)\right] \right\|_2$$
$$\leq \underbrace{\|(I - \eta_1(\lambda I + H_0))\Delta_{t-1}\|_2}_{I_1} + \underbrace{\eta_1\|f_{t-1} - y_h\|_2\|G_{t-1} - G_0\|_2}_{I_2} + \underbrace{\eta_1\|G_0\|_2\|f_{t-1} - G_0^T(\theta_{h,t-1} - \theta_0)\|_2}_{I_3}.$$

We bound $I_1$, $I_2$ and $I_3$ separately.

**Bounding $I_1$.** By Lemma A.1, we have

$$\|H_0\|_2 \leq \|G_0\|_2^2 \leq K'C_g^2.$$

Now we choose $\eta_1$ such that

$$\eta_1(\lambda + K'C_g^2) \leq 1.$$

With this choice of $\eta_1$, we have

$$I_1 = \|(I - \eta_1(\lambda I + H_0))\Delta_{t-1}\|_2 \leq \|I - \eta_1(\lambda I + H_0)\|_2 \cdot \|\Delta_{t-1}\|_2 \leq (1 - \eta_1\lambda)\|\Delta_{t-1}\|_2,$$

since $\eta_1\lambda I \preceq \eta_1(\lambda I + H_0) \preceq \eta_1(\lambda + \|G_0\|^2)I \preceq \eta_1(\lambda + K'C_g^2)I \preceq I$.

**Bounding $I_2$.** We have,

$$I_2 = \eta_1\|f_{t-1} - y_h\|_2 \cdot \|G_{t-1} - G_0\|_2$$
$$\leq \eta_1 \underbrace{\|f_{t-1} - y_h\|_2}_{\text{Lemma C.2}} \cdot \max_k \sqrt{K'} \underbrace{\|g(x_h^k; \theta_{h,t-1}) - g(x_h^k; \theta_0)\|_2}_{\text{approx error, Lemma A.1}},$$

where the inequality holds due to Cauchy-Schwarz inequality, and we can apply Lemma C.2 and Lemma A.1 for the first and second term, respectively, due to the induction assumption.

**Bounding $I_3$.** For bounding $I_3$,

$$I_3 = \eta_1\|G_0\|_2 \cdot \|f_{t-1} - G_0^T(\theta_{h,t-1} - \theta_0)\|_2$$
$$\leq \eta_1\sqrt{K'}C_g\sqrt{K'} \max_k |f(x_h^k; \widetilde{W}_{h,j}^i) - g(x_h^k; W_0)^T(\widetilde{W}_{h,j}^i - W_0)|$$
$$\leq \eta_1 K'C_g R^{4/3}m^{-1/6}\sqrt{\ln m},$$

where the first inequality holds due to Cauchy-Schwarz inequality and due to that $\|G_{h,0}\|_2 \leq \sqrt{K'}C_g$ and the second inequality holds due to the induction assumption and Lemma A.1. Combining the bounds of $I_1, I_2, I_3$ above, we have

$$\|\Delta_t\|_2 \leq (1 - \eta_1\lambda)\|\Delta_{t-1}\|_2 + I_2 + I_3.$$

Recursively applying the inequality above for all $j$, we have

$$\|\Delta_t\|_2 \leq \frac{I_2 + I_3}{\eta_1\lambda}. \tag{7}$$

We have

$$\lambda\|\bar{\theta}_{h,t}^{lin} - \theta_0\|_2^2 \leq \bar{\mathcal{L}}_h(\bar{\theta}_{h,t}^{lin}) \leq \bar{\mathcal{L}}_h(\theta_0) = \|y_h\|_2^2 \leq K'(H - h + 1 + (H - h)\iota)^2,$$

where the first inequality follows from the definition of $\bar{\mathcal{L}}_h$ and the second inequality follows from that $\{\bar{\mathcal{L}}_h(\bar{\theta}_{h,t}^{lin})\}_t$ is a non-increasing sequence due to the gradient update to linear regression. Thus, we have

$$\|\bar{\theta}_{h,t}^{lin} - \theta_0\|_2 \leq \lambda^{-1}\sqrt{K'}H(1 + \iota).$$

We have

$$\|\theta_{h,t} - \theta_0\|_2 \leq \|\theta_{h,t} - \bar{\theta}_{h,t}^{lin}\|_2 + \|\bar{\theta}_{h,t}^{lin} - \theta_0\|_2$$
$$\leq \frac{I_2 + I_3}{\eta_1 \lambda} + \lambda^{-1}\sqrt{K'}H(1 + \iota).$$

$\square$

The following lemma says that under certain conditions, the network output $f_t$ trained by GD tends to stay close to its regression target $y_h$.

**Lemma C.2.** *Let* $m = \Omega\left(d^{3/2}R^{-1}\ln^{3/2}(\sqrt{m}/R)\right)$ *and* $R = \mathcal{O}\left(m^{1/2}\ln^{-3}m\right)$. *Suppose that* $\eta_1(K'C_g^2 + \lambda/2) \leq 1/2$. *Consider any* $t \in [T_1]$. *Suppose that* $\theta_{h,l} \in \mathcal{B}(\theta_0; R), \forall l \in [t]$. *Then, with probability at least* $1 - m^{-2}$, *we have*

$$\|f_t - y_h\|_2 \lesssim K'(H - h + 1)^2(1 + \iota)^2 + (\eta_1\lambda)^{-1}R^{8/3}m^{-1/3}\ln m.$$

*Proof of Lemma C.2.* Consider the rest of the proof under event $E\{\text{Lemma A.1}\}$. Let

$$e_t = f_t - f_{t-1} - G_{t-1}^T(\theta_t - \theta_{t-1}),$$
$$e_\theta = f_\theta - f_{t-1} - G_\theta^T(\theta - \theta_{t-1}).$$

Since $\|\cdot\|_2^2$ is 1-smooth, we have

$$\mathcal{L}(\theta_t) - \mathcal{L}(\theta_{t-1}) = \|f_t - y_h\|_2^2 - \|f_{t-1} - y_h\|_2^2 + \lambda\|\theta_t - \theta_0\| - \lambda\|\theta_{t-1} - \theta_0\|_2^2$$

$$\leq 2(f_{t-1} - y_h)^T(f_t - f_{t-1}) + \frac{1}{2}\|f_t - f_{t-1}\|_2^2 + 2\lambda(\theta_{t-1} - \theta_0)^T(\theta_t - \theta_{t-1}) + \frac{\lambda}{2}\|\theta_t - \theta_{t-1}\|_2^2$$

$$\leq 2(f_{t-1} - y_h)^T(e_t + G_{t-1}^T(\theta_t - \theta_{t-1})) + \frac{1}{2}\|e_t + G_{t-1}^T(\theta_t - \theta_{t-1})\|_2^2$$

$$+ 2\lambda(\theta_{t-1} - \theta_0)^T(\theta_t - \theta_{t-1}) + \frac{\lambda}{2}\|\theta_t - \theta_{t-1}\|_2^2$$

$$\leq \nabla\mathcal{L}(\theta_{t-1})^T(\theta_t - \theta_{t-1}) + 2(f_{t-1} - y_h)^Te_t + \|e_t\|_2^2 + \|G_{t-1}^T(\theta_t - \theta_{t-1})\|_2^2 + \frac{\lambda}{2}\|\theta_t - \theta_{t-1}\|_2^2$$

$$\leq -\eta_1(1 - \eta_1(K'C_g^2 + \lambda/2))\|\nabla\mathcal{L}(\theta_{t-1})\|_2^2 + 2(f_{t-1} - y_h)^Te_t + \|e_t\|_2^2$$

Since $\|\cdot\|_2^2$ is 1-strongly convex, for any $\theta$, we have

$$\mathcal{L}(\theta) - \mathcal{L}(\theta_{t-1}) \geq 2(f_{t-1} - y_h)^T(f_\theta - f_{t-1}) + 2\lambda(\theta_{t-1} - \theta_0)^T(\theta - \theta_{t-1}) + \frac{\lambda}{2}\|\theta - \theta_{t-1}\|_2^2$$

$$= \nabla\mathcal{L}(\theta_{t-1})^T(\theta - \theta_{t-1}) + 2(f_{t-1} - y_h)^Te_\theta + \frac{\lambda}{2}\|\theta - \theta_{t-1}\|_2^2$$

$$\geq -\frac{1}{2\lambda}\|\nabla\mathcal{L}(\theta_{t-1})\|_2^2 - 2\|f_{t-1} - y_h\|_2 \cdot \|e_\theta\|_2,$$

where the last inequality uses the definition of the gradient update of $\theta_t$, and Lemma A.1 which is valid due to the condition that $\theta_{h,j} \in \mathbb{B}(\theta_0; R)$, for all $l \in [t]$. Thus, we have

$$\mathcal{L}(\theta_t) - \mathcal{L}(\theta_{t-1}) \leq \underbrace{2\eta_1\lambda(1 - \eta_1(K'C_g^2 + \lambda/2))}_{\alpha}\left(\mathcal{L}(\theta) - \mathcal{L}(\theta_{t-1}) + 2\|f_{t-1} - y_h\|_2 \cdot \|e_\theta\|_2\right)$$

$$+ 2(f_{t-1} - y_h)^Te_t + \|e_t\|_2^2$$

$$\leq \alpha\left(\mathcal{L}(\theta) - \mathcal{L}(\theta_{t-1}) + \gamma_1\|f_{t-1} - y_h\|_2^2 + \frac{1}{\gamma_1}\|e_\theta\|_2^2\right) + \gamma_2\|f_{t-1} - y_h\|_2^2 + \frac{1}{\gamma_2}\|e_t\|_2^2 + \|e_t\|_2^2$$

$$\leq \alpha\left(\mathcal{L}(\theta) - \mathcal{L}(\theta_{t-1}) + \gamma_1\mathcal{L}(\theta_{t-1}) + \frac{1}{\gamma_1}\|e_\theta\|_2^2\right) + \gamma_2\mathcal{L}(\theta_{t-1}) + \frac{1}{\gamma_2}\|e_t\|_2^2 + \|e_t\|_2^2,$$

where the second inequality uses Cauchy-Schwarz inequality for any $\gamma_1, \gamma_2 > 0$, and the third inequality uses the fact that $\|f_{t-1} - y_h\|_2^2 \leq \mathcal{L}(\theta_{t-1})$. Set $\theta = \theta_0$, and let $e = \max_{l \in [t]} \|e_l\|_2 \vee \|e_{\theta_0}\|_2$ in the above equation, we have

$$\mathcal{L}(\theta_t) \leq (1 - \alpha + \alpha\gamma_1 + \gamma_2) \mathcal{L}(\theta_{t-1}) + \left(\frac{\alpha}{\gamma_1} + \frac{1}{\gamma_2} + 1\right) e^2 + \alpha\mathcal{L}(\theta_0).$$

Now we further set that $\gamma_1 = \frac{1}{4}$ and $\gamma_2 = \frac{\alpha}{4}$. Then we have

$$\mathcal{L}(\theta_t) - \mathcal{L}(\theta_0) \leq \left(1 - \frac{\alpha}{2}\right)(\mathcal{L}(\theta_{t-1}) - \mathcal{L}(\theta_0)) + \left(4\alpha + \frac{4}{\alpha} + 1\right) e^2 + \alpha\mathcal{L}(\theta_0)/2.$$

Now we choose $\eta_1$ such that $\eta_1\lambda < \alpha < 2$ which is satisfied if we choose

$$\begin{cases} \eta_1\lambda < 1 \\ 1 - \eta_1(K'C_g^2 + \lambda/2) \geq 1/2 \end{cases}$$

Unrolling the recursion above yields

$$\mathcal{L}(\theta_t) - \mathcal{L}(\theta_0) \leq \frac{2}{\alpha}\left(\left(4\alpha + \frac{4}{\alpha} + 1\right)e^2 + \alpha\mathcal{L}(\theta_0)/2\right)$$
$$= (8 + \frac{8}{\alpha^2} + \frac{2}{\alpha})e^2 + \mathcal{L}(\theta_0).$$

Thus, we have

$$\|f_t - y_h\|_2 \leq \mathcal{L}(\theta_t) \leq 2\mathcal{L}(\theta_0) + (8 + \frac{8}{\alpha^2} + \frac{2}{\alpha})e^2$$
$$\lesssim K'(H - h + 1)^2(1 + \iota)^2 + (\eta_1\lambda)^{-1}R^{8/3}m^{-1/3}\ln m,$$

where the last inequality holds due to that $\mathcal{L}(\theta_0) = \|y_h\|_2^2 \leq K(H - h + 1)^2(1 + \iota)^2$ and $e \lesssim R^{4/3}m^{-1/6}\sqrt{\ln m}$ with probability at least $1 - m^{-2}$ (due to $\theta_{h,l} \in \mathcal{B}(\theta_0, R), \forall l \in [t]$ and Lemma A.1). $\qquad\square$

The following lemma says that under certain conditions, the weighted version of the gradient of the network output $\|g(x; \theta_h)\|_{\Lambda_h^{-1}}$ tends to stay close to its counterpart $\|g(x; \theta_0)\|_{\bar{\Lambda}_h^{-1}}$ at the initialization.

**Lemma C.3.** *Let*

$$m = \Omega\left(d^{3/2}R^{-1}\ln^{3/2}(\sqrt{m}/R)\right),$$
$$R = \mathcal{O}\left(m^{1/2}\ln^{-3}m\right),$$
$$(\eta_1\lambda)^{-1}R^{8/3}m^{-1/3}\ln m \lesssim 1,$$
$$\eta_1(K'C_g^2 + \lambda/2) \leq 1/2,$$
$$\lambda^{-1}(K'(H - h + 1)^2(1 + \iota)^2 + 1)\sqrt{K'}R^{1/3}m^{-1/6}\sqrt{m}$$
$$+ \lambda^{-1}K'C_gR^{4/3}m^{-1/6}\sqrt{\ln m} + \lambda^{-1}\sqrt{K'}H(1 + \iota) \lesssim R.$$

*With probability at least $1 - m^{-2}$, for any $x \in \mathbb{S}_{d-1}$ and $h \in [H]$, we have*

$$\left|\|g(x; \theta_h)\|_{\Lambda_h^{-1}} - \|g(x; \theta_0)\|_{\bar{\Lambda}_h^{-1}}\right| \lesssim \lambda^{-1}\sqrt{K'}R^{1/6}m^{-1/12}\ln^{1/4}m + \lambda^{-1/2}R^{1/3}m^{-1/6}\sqrt{\ln m},$$

*where $\theta_h = \theta_{h,T_1}$ which is returned after phase I of Algorithm 3.*

*Proof of Lemma C.3.* For simplicity, let us define

$$G_1 = \{g(x_h^k; \hat{\theta}_h)\}_{k \in \mathcal{I}_h} \in \mathbb{R}^{dm \times K'} \text{ and } G_0 = \{g(x_h^k; \theta_0)\}_{k \in \mathcal{I}_h} \in \mathbb{R}^{dm \times K'}.$$

By Lemma C.1, we have $\theta_h \in \mathcal{B}(\theta_0; R)$. Under the joint event $E\{$Lemma C.1$\} \cap E\{$Lemma A.1$\}$, for any $x$ and $h$, we have

$$\|g(x;\theta_h)\|_{\Lambda_h^{-1}} - \|g(x;\theta_0)\|_{\bar{\Lambda}_h^{-1}} \leq \|g(x;\theta_h)\|_2 \sqrt{\|\Lambda_h^{-1} - \bar{\Lambda}_h^{-1}\|_2} + \|g(x;\theta_h)\|_{\bar{\Lambda}_h^{-1}} - \|g(x;\theta_0)\|_{\bar{\Lambda}_h^{-1}}$$

$$\leq \|g(x;\theta_h)\|_2 \sqrt{\|\Lambda_h^{-1} - \bar{\Lambda}_h^{-1}\|_2} + \|g(x;\theta_h) - g(x;\theta_0)\|_{\bar{\Lambda}_h^{-1}}$$

$$\leq \|g(x;\theta_h)\|_2 \sqrt{\|\Lambda_h^{-1} - \bar{\Lambda}_h^{-1}\|_2} + \|g(x;\theta_h) - g(x;\theta_0)\|_2 \sqrt{\|\bar{\Lambda}_h^{-1}\|_2}$$

$$\leq \|g(x;\theta_h)\|_2 \sqrt{\frac{\|\Lambda_h - \bar{\Lambda}_h\|_2}{\lambda_{\min}(\Lambda_h)\lambda_{\min}(\bar{\Lambda}_h)}} + \|g(x;\theta_h) - g(x;\theta_0)\|_2 \sqrt{\|\bar{\Lambda}_h^{-1}\|_2}$$

$$\leq C_g \sqrt{2} \sqrt{K'} \sqrt{C_g} R^{1/6} m^{-1/12} \ln^{1/4} m \cdot \lambda^{-1} + R^{1/3} m^{-1/6} \sqrt{\ln m} \lambda^{-1/2}$$

where the fourth inequality follows from Lemma A.8, and the last inequality follows from $\theta_h \in \mathbb{B}(\theta_0; R)$ and Lemma A.1, and

$$\begin{aligned}
\|\Lambda_h - \bar{\Lambda}_h\|_2 &= \|G_1 G_1^T - G_0 G_0^T\|_2 \\
&= \|G_1(G_1 - G_0)^T + (G_1 - G_0)G_0^T\|_2 \\
&\leq \|G_1\|_2 \|G_1 - G_0\|_2 + \|G_1 - G_0\|_2 \|G_0\|_2 \\
&\leq 2\sqrt{K'} C_g \sqrt{K'} R^{1/3} m^{-1/6} \sqrt{m}.
\end{aligned}$$

$\square$

## C.3 APPROXIMATE POSTERIOR SAMPLING FROM LANGEVIN DYNAMICS

In this part, we show that our algorithm performs an approximate posterior sampling from the data posterior distribution. For simplicity, we denote $\theta_h^{lin,i} = \theta_{h,T_2}^{lin,i}$ which is returned by the LMC to the auxiliary linear model (after Line 14 of Algorithm 3). We also denote $\theta_h^{lin} = \theta_{h,T_2}^{lin}$ which is returned by the gradient descent to the auxiliary linear model (after Line 10 of Algorithm 3). Similar to Lemma B.1, the following lemma shows that the parameter samples $\{\theta_h^{lin,i}\}_{i\in[M]}$ from Langevin dynamics approximate the samples of the data posterior.

**Lemma C.4.** *For any $h \in [H]$, we have*

$$\{\theta_h^{lin,i}\}_{i\in[M]} \overset{i.i.d.}{\sim} \mathcal{N}(\theta_h^{lin}, \Sigma_h)$$

*where*

$$\Sigma_h := \tau(I - A_h^{2T_2})\Lambda_h^{-1}(I + A_h)^{-1}, \quad A_h := I - 2\eta_2 \Lambda_h,$$

*In addition, we have $\theta_h^{lin} = A_h^{T_2}\theta_0 + (I - A_h^{T_2})\hat{\theta}_h^{lin}$. Furthermore, if we set*

$$\eta_2 = \frac{1}{4\lambda_{\max}(\Lambda_h)}$$

*then, we have*

$$\left(1 - \left(1 - \frac{1}{2\kappa(\Lambda_h)}\right)^{T_2}\right)\tau\Lambda_h^{-1} \preceq \Sigma_h \preceq \tau\Lambda_h^{-1}. \tag{8}$$

*Proof of Lemma C.4.* Unroll the recursion in the gradient descent update in Line 9 of Algorithm 3, we have

$$\begin{aligned}
\theta_{h,t}^{lin,i} &= \theta_{h,t-1}^{lin,i} - \eta_2 \nabla \mathcal{L}_h^{lin}(\theta_{h,t-1}^{lin,i}) + \sqrt{2\eta\tau}\epsilon_t \\
&= \theta_{h,t-1}^{lin,i} - 2\eta_2 \left(\Lambda_h \theta_{h,t-1}^{lin,i} - G_h(\theta_h)^T y_h - \lambda\theta_0\right) + \sqrt{2\eta\tau}\epsilon_t \\
&= A_h \theta_{h,t-1}^{lin,i} + 2\eta_2 \left(G_h(\theta_h)^T y_h + \lambda\theta_0\right) + \sqrt{2\eta\tau}\epsilon_t
\end{aligned}$$

$$= A_h^t \theta_0 + 2\eta_2 \sum_{l=0}^{t-1} A_h^l \left( G_h(\hat{\theta}_h)^T y_h + \lambda \theta_0 \right) + \sqrt{\eta\tau} \sum_{l=0}^{t-1} A_h^l \epsilon_{t-l}$$

$$= A_h^t \theta_0 + 2\eta_2 (I - A_h^t)(I + A_h)^{-1}(I - A_h)^{-1} \left( G_h(\hat{\theta}_h)^T y_h + \lambda \theta_0 \right) + \sqrt{2\eta\tau} \sum_{l=0}^{t-1} A_h^l \epsilon_{t-l}$$

$$= A_h^t \theta_0 + (I - A_h^t)\hat{\theta}_h^{lin} + \sqrt{2\eta\tau} \sum_{l=0}^{t-1} A_h^l \epsilon_{t-l}.$$

Since $\{\epsilon_t\}_{t \in [T_2]}$ are mutually independent Gaussian noises, we have

$$\theta_{h,T_2}^{lin,i} \sim \mathcal{N}(\nu_h, \Sigma_h),$$

where $\nu_h = A_h^{T_2}\theta_0 + (I - A_h^{T_2})\hat{\theta}_h^{lin}$, and

$$\Sigma_h = 2\eta_2 \tau \sum_{t=0}^{T_2-1} A_h^{2t} = 2\eta_2 \tau (I - A^{2T_2})(I - A_h)^{-1}(I + A_h)^{-1} = \tau(I - A^{2T_2})\Lambda_h^{-1}(I + A_h)^{-1}.$$

Note that $\theta_h^{lin} = \text{GD}(\mathcal{L}_h^{lin}, \theta_0, \eta_2, T_2) = \text{LGD}(\mathcal{L}_h^{lin}, \theta_0, \eta_2, T_2, \tau = 0)$, we have $\theta_h^{lin} = \nu_h$.

Now we set $\eta_2$ such that

$$\eta_2 = \frac{1}{4\lambda_{\max}(\Lambda_h)}$$

We define the condition number:

$$\kappa_h := \frac{\lambda_{\max}(\Lambda_h)}{\lambda_{\min}(\Lambda_h)}$$

$$0 < \eta_2 \leq \frac{1}{2\lambda_{\max}(\Lambda_h)} \wedge \frac{1}{2\lambda}.$$

With this choice of $\eta_2$, we have

$$\begin{cases} 0 & \preceq A_h = I - 2\eta_2\Lambda_h \preceq (1 - 2\eta_2\lambda_{\min}(\Lambda_h))I = (1 - 1/(2\kappa_h))I, \\ I & \preceq I + A_h \preceq 2I, \\ I & \succ I - A_h^{2T_2} \succeq (1 - (1 - (2\kappa_h)^{-1})^{T_2})I. \end{cases} \tag{9}$$

Thus, we have

$$(1 - (1 - \frac{1}{2\kappa_h})^{T_2})\tau\Lambda_h^{-1} \preceq \Sigma_h \preceq \tau\Lambda_h^{-1}.$$

$\square$

## C.4 PESSIMISM

In this part, we show how pessimism is obtained in our approximate posterior sampling framework. This result is formally stated in the following lemma.

**Lemma C.5.** *Let*

$$m = \Omega\left(d^{3/2}R^{-1}\ln^{3/2}(\sqrt{m}/R)\right),$$

$$R = \mathcal{O}\left(m^{1/2}\ln^{-3}m\right),$$

$$(\eta_1\lambda)^{-1}R^{8/3}m^{-1/3}\ln m \lesssim 1,$$

$$\eta_1(K'C_g^2 + \lambda/2) \leq 1/2,$$

$$T_2 \geq \frac{\ln 2}{\ln(1/(1 - 2\eta\lambda))},$$

$$\sqrt{\tau} \geq 2\gamma$$

$$M = \ln \frac{H|\mathcal{S}|}{\delta} \Big/ \ln \frac{1}{1 - \Phi(-1)},$$

$$\lambda^{-1}(K'(H - h + 1)^2(1 + \iota)^2 + 1)\sqrt{K'}R^{1/3}m^{-1/6}\sqrt{m}$$
$$+ \lambda^{-1}K'C_g R^{4/3}m^{-1/6}\sqrt{\ln m} + \lambda^{-1}\sqrt{K'}H(1 + \iota) \lesssim R,$$

*where $\gamma$ is defined in Lemma C.8. For any deterministic policy $\pi$, with probability at least $1 - m^{-2} - 2\delta$, for all $s \in \mathcal{S}$ and all $h \in [H]$, we have*

$$\text{err}_h(s, \pi_h(s)) \geq -R^{4/3}m^{-1/6}\sqrt{\ln m} - \lambda^{-1}(K'H^2(1 + \iota)^2 + 1)\sqrt{K'}R^{1/3}m^{-1/6}\sqrt{m}$$
$$+ \lambda^{-1}K'R^{4/3}m^{-1/6}\sqrt{\ln m} - (1 - 2\eta\lambda_{\max}(\bar{\Lambda}_h))^{T_1}\lambda^{-1/2}K'H(1 + \iota)$$
$$- \gamma\lambda^{-1}\sqrt{K'}R^{1/6}m^{-1/12}\ln^{1/4}m - \gamma\lambda^{-1/2}R^{1/3}m^{-1/6}\sqrt{\ln m}.$$

*Proof of Lemma C.5.* Consider the joint even $E\{\text{Lemma C.7}\} \cap E\{\text{Lemma C.7}\} \cap E\{\text{Lemma C.8}\}$. For any $x$, if $f(x; \theta_h) + \min_{i \in [M]}\langle g(x; \theta_h), \theta_h^{lin,i} - \theta_h^{lin}\rangle < 0$, then $\widetilde{Q}_h(x) = 0$, thus $\text{err}_h(x) = \mathbb{B}_h\widetilde{V}_{h+1}(x) - \widetilde{Q}_h(x) = \mathbb{B}_h\widetilde{V}_{h+1} \geq 0$ since $r_h \geq 0$ and $\widetilde{V}_{h+1} \geq 0$. Consider the case $f(x; \theta_h) + \min_{i \in [M]}\langle g(x; \theta_h), \theta_h^{lin,i} - \theta_h^{lin}\rangle \geq 0$. Under the joint event $E\{\text{Lemma C.7}\} \cap E\{\text{Lemma C.7}\}$, for any $x$, we have

$$\widetilde{Q}_h(x) = \min\{f(x; \theta_h) + \min_{i \in [M]}\langle g(x; \theta_h), \theta_h^{lin,i} - \theta_h^{lin}\rangle, (H - h + 1)(1 + \iota)\}^+$$

$$\leq f(x; \theta_h) + \min_{i \in [M]}\langle g(x; \theta_h), \theta_h^{lin,i} - \theta_h^{lin}\rangle$$

$$\leq f(x; \theta_h) - \|g(x; \theta_h)\|_{\Sigma_h},$$

where the third inequality follows from Lemma C.7. Thus, we have

$$\text{err}_h(x) = \mathbb{B}_h\widetilde{V}_{h+1}(x) - \widetilde{Q}_h(x)$$

$$\geq \mathbb{B}_h\widetilde{V}_{h+1}(x) - f(x; \theta_h) + \|g(x; \theta_h)\|_{\Sigma_h}$$

$$\geq \langle g(x; \theta_0), \hat{\theta}_h^{lin} - \theta_0\rangle - \gamma\|g(x; \theta_0)\|_{\bar{\Lambda}_h^{-1}} - f(x; \theta_h) + \|g(x; \theta_h)\|_{\Sigma_h}$$

$$\geq \langle g(x; \theta_0), \hat{\theta}_h^{lin} - \theta_0\rangle - f(x; \theta_h) - \gamma\|g(x; \theta_0)\|_{\bar{\Lambda}_h^{-1}} + (1 - (1 - 2\eta\lambda)^{T_2})\tau\|g(x; \theta_h)\|_{\Lambda_h^{-1}}$$

$$\geq \langle g(x; \theta_0), \hat{\theta}_h^{lin} - \theta_0\rangle - f(x; \theta_h) - \gamma\|g(x; \theta_0)\|_{\bar{\Lambda}_h^{-1}} + 0.5\tau\|g(x; \theta_h)\|_{\Lambda_h^{-1}}$$

$$= \underbrace{\langle g(x; \theta_0), \hat{\theta}_h^{lin} - \theta_0\rangle - f(x; \theta_h)}_{\text{approx + opt error, Lemma C.7}} + \underbrace{(0.5\sqrt{\tau} - \gamma)}_{\geq 0, \text{ by choice of } \tau}\|g(x; \theta_0)\|_{\bar{\Lambda}_h^{-1}}$$

$$+ 0.5\sqrt{\tau}\underbrace{\left(\|g(x; \theta_h)\|_{\Lambda_h^{-1}} - \|g(x; \theta_0)\|_{\bar{\Lambda}_h^{-1}}\right)}_{\text{Lemma C.3}}$$

where the second inequality follows from Lemma C.8, the third inequality follows from Equation (8), the fourth inequality follows from the choice of $T_2$. Applying Lemma C.7 and Lemma C.3 to the last inequality above completes our proof. □

The following lemma characterizes the approximation error and the optimization error of a linear model $\langle g(x; \theta_0), \hat{\theta}_h^{lin} - \theta_0\rangle$ constructed from the neural tangent features to the network output $f(x; \theta_h)$ trained by GD.

**Lemma C.6.** *Let*

$$m = \Omega\left(d^{3/2}R^{-1}\ln^{3/2}(\sqrt{m}/R)\right),$$

$$R = \mathcal{O}\left(m^{1/2}\ln^{-3}m\right),$$

$$(\eta_1\lambda)^{-1}R^{8/3}m^{-1/3}\ln m \lesssim 1,$$

$$\eta_1 \leq \frac{1}{4\lambda_{\max}(\bar{\Lambda}_h)},$$

$$\lambda^{-1}(K'(H-h+1)^2(1+\iota)^2+1)\sqrt{K'}R^{1/3}m^{-1/6}\sqrt{m}$$

$$+ \lambda^{-1}K'C_gR^{4/3}m^{-1/6}\sqrt{\ln m} + \lambda^{-1}\sqrt{K'}H(1+\iota) \lesssim R.$$

*With probability at least $1 - m^{-2}$, for any $x \in \mathbb{S}_{d-1}$ and any $h \in [H]$, we have*

$$|f(x;\theta_h) - \langle g(x;\theta_0), \hat{\bar{\theta}}_h^{lin} - \theta_0 \rangle| \lesssim R^{4/3}m^{-1/6}\sqrt{\ln m}$$

$$+ \lambda^{-1}(K'H^2(1+\iota)^2+1)\sqrt{K}R^{1/3}m^{-1/6}\sqrt{m} + \lambda^{-1}K'R^{4/3}m^{-1/6}\sqrt{\ln m}$$

$$+ (1 - 2\eta_1\lambda_{\min}(\bar{\Lambda}_h))^{T_1}\lambda^{-1/2}K'H(1+\iota)$$

*where $\hat{\bar{\theta}}_h^{lin} = \bar{\Lambda}_h^{-1}\left(G(\theta_0)y_h + \lambda\theta_0\right)$ defined in [Equation (6)](#).*

*Proof of [Lemma C.6](#).* We have

$$|f(x;\theta_h) - \langle g(x;\theta_0), \hat{\bar{\theta}}_h^{lin} - \theta_0 \rangle| \leq |f(x;\theta_h) - \langle g(x;\theta_0), \theta_h - \theta_0 \rangle| + |g(x;\theta_0)^T(\theta_h - \bar{\theta}_{h,T_1}^{lin})|$$

$$+ |\langle g(x;\theta_0), \bar{\theta}_{h,T_1}^{lin} - \hat{\bar{\theta}}_h^{lin} \rangle|$$

where $\bar{\theta}_{h,T_1}^{lin}$ is defined in [Equation (5)](#). By [Lemma C.1](#), $\theta_h \in \mathcal{B}(\theta_0; R)$, thus by [Lemma A.1](#), under the event $E\{$[Lemma A.1](#)$\}$, for any $x \in \mathbb{S}_{d-1}$, we have

$$|f(x;\theta_h) - g(x;\theta_0)^T(\theta_h - \theta_0)| \leq C_gR^{4/3}m^{-1/6}\sqrt{\ln m}.$$

By [Lemma C.1](#), under the event $E\{$[Lemma C.1](#)$\}$, we have

$$\|\theta_h - \bar{\theta}_{h,T_1}^{lin}\|_2 \lesssim \lambda^{-1}(K'(H-h+1)^2(1+\iota)^2+1)\sqrt{K'}R^{1/3}m^{-1/6}\sqrt{m} + \lambda^{-1}K'C_gR^{4/3}m^{-1/6}\sqrt{\ln m}$$

For bounding the third term, let $\bar{A}_h = I - 2\eta_1\bar{\Lambda}_h$. Under the event $E\{$[Lemma A.1](#)$\}$, we have

$$|\langle g(x;\theta_0), \bar{\theta}_{h,T_1}^{lin} - \hat{\bar{\theta}}_h^{lin} \rangle| \leq \|g(x;\theta_0)\|_2 \cdot \|\bar{\theta}_{h,T_1}^{lin} - \hat{\bar{\theta}}_h^{lin}\|_2$$

$$\leq C_g\|\bar{\theta}_{h,T_1}^{lin} - \hat{\bar{\theta}}_h^{lin}\|_2$$

$$= C_g\|\bar{A}_h^{T_1}(\theta_0 - \hat{\bar{\theta}}_h^{lin})\|_2$$

$$\leq C_g(1 - 2\eta_1\lambda_{\min}(\bar{\Lambda}_h))^{T_1}\|\theta_0 - \hat{\bar{\theta}}_h^{lin}\|_2$$

$$= C_g(1 - 2\eta_1\lambda_{\min}(\bar{\Lambda}_h))^{T_1} \cdot \|\bar{\Lambda}_h^{-1}\left(G(\theta_0)y_h + \lambda\theta_0\right) - \bar{\Lambda}_h^{-1}\bar{\Lambda}_h\theta_0\|_2$$

$$= C_g(1 - 2\eta_1\lambda_{\min}(\bar{\Lambda}_h))^{T_1} \cdot \|\bar{\Lambda}_h^{-1}\left(G(\theta_0)y_h - G(\theta_0)G(\theta_0)^T\theta_0\right)\|_2$$

$$= C_g(1 - 2\eta_1\lambda_{\min}(\bar{\Lambda}_h))^{T_1}\|\bar{\Lambda}_h^{-1}G(\theta_0)y_h\|_2$$

$$\leq C_g(1 - 2\eta_1\lambda_{\min}(\bar{\Lambda}_h))^{T_1}\lambda^{-1/2}K'C_gH(1+\iota),$$

where the second inequality follows from [Lemma A.1](#), the first equality follows from unrolling the gradient update linear regression, the third inequality follows from the definition of $\bar{\theta}_{h,T_1}^{lin}$ is defined in [Equation (5)](#), the fourth equality follows from that $g(x;\theta_0)^T\theta_0, \forall x$ due to the symmetric initialization of $\theta_0$, and the last inequality follows from $\|\bar{\Lambda}_h^{-1}\|_2 \leq \lambda^{-1}$, $\|G(\theta_0)\|_2 \leq \sqrt{K'}C_g$ due to [Lemma A.1](#), and $\|y_2\|_2 \leq \sqrt{K'}H(1+\iota)$.

Altogether, under the joint event $E\{$[Lemma C.1](#)$\} \cap E\{$[Lemma A.1](#)$\}$, we conclude our statement via the union bound. $\qquad\square$

The following lemma specifies the anti-concentration of Gaussian distributions.

**Lemma C.7.** *Let $M = \ln\frac{H|\mathcal{S}|}{\delta} / \ln\frac{1}{1-\Phi(-1)}$ where $\Phi(\cdot)$ is the cumulative distribution function of the standard normal distribution. For any deterministic policy $\pi$, with probability at least $1 - \delta$, for any $(s,h) \in \mathcal{S} \times [H]$, we have*

$$\min_{i\in[M]}\langle g(s,\pi(s);\theta_h), \theta_h^{lin,i} \rangle \leq \langle g(s,\pi(s);\theta_h), \theta_h^{lin} \rangle - \|g(s,\pi(s);\theta_h)\|_{\Sigma_h}.$$

*Proof of Lemma C.7.* We have

$$\langle g(s, \pi(s); \theta_h), \theta_h^{lin,i} - \theta_h^{lin} \rangle \sim \mathcal{N}(0, \|g(s, \pi(s); \theta_h)\|_{\Sigma_h}).$$

By the anti-concentration of Gaussian distributions, we have

$$\Pr\left(\langle g(s, \pi(s); \theta_h), \theta_h^{lin,i} - \theta_h^{lin} \rangle \leq -\|g(s, \pi(s); \theta_h)\|_{\Sigma_h}\right) = \Phi(-1).$$

Since $\{\theta_h^{lin,i}\}_{i \in [M]}$ are *mutually independent*, with probability at least $1 - (1 - \Phi(-1))^M$, we have

$$\min_{i \in [M]} \langle g(s, \pi(s); \theta_h), \theta_h^{lin,i} - \theta_h^{lin} \rangle \leq -\|g(s, \pi(s); \theta_h)\|_{\Sigma_h}.$$

We set $\delta = (1 - \Phi(-1))^M$ to complete the proof. $\qquad\square$

The following lemma characterizes the estimation error of using the linear model at its mode $\langle g(x; \theta_0), \hat{\theta}_h^{lin} - \theta_0 \rangle$ to estimate the Bellman target $(\mathbb{B}_h \widetilde{V}_{h+1})(x)$.

**Lemma C.8.** *Let $\lambda > 1, m = \Omega\left(K'^{10} H^8 (1 + \iota)^8 \ln(3K'H/\delta)\right)$. With probability at least $1 - m^{-2} - \delta$, for any $x \in \mathbb{S}_{d-1}$ and any $h \in [H]$, we have*

$$|(\mathbb{B}_h \widetilde{V}_{h+1})(x) - \langle g(x; \theta_0), \hat{\theta}_h^{lin} - \theta_0 \rangle| \leq \frac{B}{\sqrt{m}}(2\sqrt{d} + \sqrt{2\ln(3H/\delta)}) + \xi_h + \gamma \|g(x; \theta_0)\|_{\bar{\Lambda}_h^{-1}},$$

*where*

$$\gamma \lesssim B\sqrt{\lambda} + \sqrt{K'}\lambda^{-1}\left(\frac{B}{\sqrt{m}}(2\sqrt{d} + \sqrt{2\ln(3H/\delta)}) + \xi_h\right)$$
$$+ H(1 + \iota)\sqrt{\tilde{d}_h \ln(1 + K'/\lambda) + K' \ln\lambda + 2\ln(3H/\delta)} + 1.$$

*Proof of Lemma C.8.* We have

$$(\mathbb{B}_h \widetilde{V}_{h+1})(x) - \langle g(x; \theta_0), \hat{\theta}_h^{lin} - \theta_0 \rangle = \underbrace{\mathbb{B}_h \widetilde{V}_{h+1}(x) - \langle g(x; \theta_0), \theta_h^* - \theta_0 \rangle}_{\text{approx error}} + \underbrace{\langle g(x; \theta_0), \theta_h^* - \hat{\theta}_h^{lin} \rangle}_{\text{estimation error}},$$

where the first term is the approximation error and the second term is the estimation error. To bound the approximation error (the first term), under the event $E\{\text{Lemma C.9}\}$, we have

$$|\mathbb{B}_h \widetilde{V}_{h+1}(x) - \langle g(x; \theta_0), \theta_h^* - \theta_0 \rangle| \leq \frac{B}{\sqrt{m}}(2\sqrt{d} + \sqrt{2\ln(H/\delta)}) + \xi_h.$$

To bound the estimation error (the second term), we have

$$\langle g(x; \theta_0), \theta_h^* - \theta_0 \rangle - \langle g(x; \theta_0), \hat{\theta}_h^{lin} - \theta_0 \rangle = g(x; \theta_0)^T(\theta_h^* - \theta_0) - g(x; \theta_0)^T \bar{\Lambda}_h^{-1} \sum_{k \in \mathcal{I}_h} g(x_h^k; \theta_0) y_h^k$$

$$= \underbrace{g(x; \theta_0)^T(\theta_h^* - \theta_0) - g(x; \theta_0)^T \bar{\Lambda}_h^{-1} \sum_{k \in \mathcal{I}_h} g(x_h^k; \theta_0) \cdot (\mathbb{B}_h \tilde{V}_{h+1})(x_h^k)}_{I_1}$$

$$+ \underbrace{g(x; \theta_0)^T \bar{\Lambda}_h^{-1} \sum_{k \in \mathcal{I}_h} g(x_h^k; \theta_0) \cdot \left[(\mathbb{B}_h \tilde{V}_{h+1})(x_h^k) - (r_h^k + \tilde{V}_{h+1}(s_{h+1}^k))\right]}_{I_2}.$$

**Bounding term $I_1$:** Under the joint event $E\{\text{Lemma C.9}\} \cap E\{\text{Lemma A.1}\}$, we have

$$|I_1| = |g(x; \theta_0)^T(\theta_h^* - \theta_0) - g(x; \theta_0)^T \bar{\Lambda}_h^{-1} \sum_{k \in \mathcal{I}_h} g(x_h^k; \theta_0) \cdot (\mathbb{B}_h \tilde{V}_{h+1})(x_h^k)|$$

$$= |\lambda g(x; \theta_0)\bar{\Lambda}_h^{-1}(\theta^* - \theta_0) - g(x; \theta_0)^T \bar{\Lambda}_h^{-1} \sum_{k \in \mathcal{I}_h} g(x_h^k; \theta_0) \cdot \left((\mathbb{B}_h \tilde{V}_{h+1})(x_h^k) - g(x_h^k; \theta_0)^T(\theta_h^* - \theta_0)\right)|$$

$$\leq \lambda \|g(x;\theta_0)\|_{\bar{\Lambda}_h^{-1}} \cdot \|\theta^* - \theta_0\|_{\bar{\Lambda}_h^{-1}}$$

$$+ \|g(x;\theta_0)\|_{\bar{\Lambda}_h^{-1}} \cdot \| \sum_{k \in \mathcal{I}_h} g(x_h^k;\theta_0) \cdot \left( (\mathbb{B}_h \tilde{V}_{h+1})(x_h^k) - g(x_h^k;\theta_0)^T(\theta_h^* - \theta_0) \right) \|_{\bar{\Lambda}_h^{-1}}$$

$$\leq \|g(x;\theta_0)\|_{\bar{\Lambda}_h^{-1}} \left( \lambda B \lambda^{-1/2} + \sqrt{K'} C_g^2 \lambda^{-1}(\frac{B}{\sqrt{m}}(2\sqrt{d} + \sqrt{2\ln(H/\delta)}) + \xi_h) \right),$$

where the second equality follows from the definition of $\bar{\Lambda}_h$, the first inequality follows from the triangle inequality and that $x^T A y \leq \|x^T A^{1/2}\|_2 \|A^{1/2}y\|_2 = \|x\|_A \|y\|_A$, and the second inequality follows the inequality $\|x\|_A \leq \sqrt{\|A\|_2}\|x\|_2$, $\|\bar{\Lambda}_h\|_2 \leq \lambda^{-1}$, Lemma C.9, and $\|g(x;\theta_0)\|_2 \leq C_g$ by Lemma A.1.

**Bounding term $I_2$.** We have

$$|I_2| = |g(x;\theta_0)^T \bar{\Lambda}_h^{-1} \sum_{k \in \mathcal{I}_h} g(x_h^k;\theta_0) \cdot \left[ (\mathbb{B}_h \tilde{V}_{h+1})(x_h^k) - (r_h^k + \tilde{V}_{h+1}(s_{h+1}^k)) \right]|$$

$$\leq \|g(x;\theta_0)\|_{\bar{\Lambda}_h^{-1}} \cdot \underbrace{\| \sum_{k \in \mathcal{I}_h} g(x_h^k;\theta_0) \cdot \left[ (\mathbb{B}_h \tilde{V}_{h+1})(x_h^k) - (r_h^k + \tilde{V}_{h+1}(s_{h+1}^k)) \right] \|_{\bar{\Lambda}_h^{-1}}}_{I_3} .$$

For notational simplicity, we write

$$\epsilon_h^k := (\mathbb{B}_h \tilde{V}_{h+1})(x_h^k) - r_h^k - \tilde{V}_{h+1}(s_{h+1}^k),$$
$$E_h := [(\epsilon_h^k)_{k \in \mathcal{I}_h}]^T \in \mathbb{R}^{K'}.$$

We denote $\mathcal{K}_h^{init} := [\langle g(x_h^i;\theta_0), g(x_h^j;\theta_0) \rangle]_{i,j \in \mathcal{I}_h}$ as the Gram matrix of the empirical NTK kernel on the data $\{x_h^k\}_{k \in [K]}$. We denote

$$G_0 := \left( g(x_h^k;\theta_0) \right)_{k \in \mathcal{I}_h} \in \mathbb{R}^{md \times K'},$$
$$\mathcal{K}_h^{int} := G_0^T G_0 \in \mathcal{R}^{K' \times K'}.$$

Recall the definition of the Gram matrix $\mathcal{K}_h$ of the NTK kernel on the data $\{x_h^k\}_{k \in \mathcal{I}_h}$. It follows from Lemma A.2 and the union bound that if $m = \Omega(\epsilon^{-4} \ln(3K'H/\delta))$ with probability at least $1 - \delta/3$, for any $h \in [H]$,

$$\|\mathcal{K}_h - \mathcal{K}_h^{init}\|_F \leq \sqrt{K'}\epsilon. \tag{10}$$

**We now can bound $I_3$.** We have

$$I_3^2 = \left\| \sum_{k \in \mathcal{I}_h} g(x_h^k;\theta_0)\epsilon_h^k \right\|_{\bar{\Lambda}_h^{-1}}^2$$

$$= E_h^T G_0^T (\lambda I_{md} + G_0 G_0^T)^{-1} G_0 E_h$$

$$= E_h^T G_0^T G_0 (\lambda I_{K'} + G_0^T G_0)^{-1} E_h$$

$$= E_h^T \mathcal{K}_h^{init} (\mathcal{K}_h^{init} + \lambda I_K)^{-1} E_h$$

$$= \underbrace{E_h^T \mathcal{K}_h (\mathcal{K}_h + \lambda I_{K'})^{-1} E_h}_{I_5, \text{estimation error}} + \underbrace{E_h^T \left( \mathcal{K}_h (\mathcal{K}_h + \lambda I_{K'})^{-1} - \mathcal{K}_h^{init}(\mathcal{K}_h^{int} + \lambda I_{K'})^{-1} \right) E_h}_{I_4, \text{approx. error}}. \tag{11}$$

**For bounding $I_4$,** consider the joint event $E\{\text{Lemma A.1}\} \cap E\{\text{Equation (10)}\}$. Under this joint event, we have

$$I_4 \leq \left\| \mathcal{K}_h (\mathcal{K}_h + \lambda I_{K'})^{-1} - \mathcal{K}_h^{init}(\mathcal{K}_h^{int} + \lambda I_{K'})^{-1} \right\|_2 \|E_h\|_2^2$$

$$= \left\| (\mathcal{K}_h - \mathcal{K}_h^{init})(\mathcal{K}_h + \lambda I_{K'})^{-1} + \mathcal{K}_h^{init} \left( (\mathcal{K}_h + \lambda I_{K'})^{-1} - (\mathcal{K}_h^{int} + \lambda I_{K'})^{-1} \right) \right\|_2 \|E_h\|_2^2$$

$$\leq \|\mathcal{K}_h - \mathcal{K}_h^{init}\|_2/\lambda + \|\mathcal{K}_h^{init}\|_2 \cdot \|\mathcal{K}_h - \mathcal{K}_h^{init}\|_2/\lambda^2 \|E_h\|_2^2$$

$$\leq \frac{\lambda + K'C_g^2}{\lambda^2} \|\mathcal{K}_h - \mathcal{K}_h^{init}\|_2 \|E_h\|_2^2$$

$$\leq \frac{\lambda + K'C_g^2}{\lambda^2} K'H^2(1+\iota)^2\sqrt{K'}\epsilon$$
$$= 1 \tag{12}$$

where the first inequality holds due to $\|x\|_A \leq \sqrt{\|A\|_2}\|x\|_2$, the second inequality holds due to the triangle inequality, Lemma A.8, and $\|(\mathcal{K}_h + \lambda I_{K'})^{-1}\|_2 \leq \lambda^{-1}$, the third inequality holds due to $\|\mathcal{K}_h^{init}\|_2 \leq \|G_0\|_2^2 \leq \|G_0\|_F^2 \leq K'C_g^2$ due to Lemma A.1, the fourth inequality holds due to $\|E_h\|_2 \leq \sqrt{K'}H(1+\iota)$, and Equation (10), and the last inequality holds if we choose $\frac{1}{\epsilon} = \frac{\lambda + K'C_g^2}{\lambda^2}K'H^2(1+\iota)^2\sqrt{K'}$ in Equation (10). This choice of $\epsilon$ leads to the condition:

$$m = \Omega\left(K^{10}H^8(1+\iota)^8\ln(3K'H/\delta)\right). \tag{13}$$

**For bounding** $I_5$, as $\lambda > 1$, we have

$$I_5 = E_h^T\mathcal{K}_h(\mathcal{K}_h + \lambda I_{K'})^{-1}E_h$$
$$\leq E_h^T(\mathcal{K}_h + (\lambda - 1)I_K)(\mathcal{K}_h + \lambda I_{K'})^{-1}E_h$$
$$= E_h^T\left[(\mathcal{K}_h + (\lambda - 1)I_{K'})^{-1} + I_{K'}\right]^{-1}E_h. \tag{14}$$

Let $\sigma(\cdot)$ be the $\sigma$-algebra induced by the set of random variables. For any $h \in [H]$ and $k \in \mathcal{I}_h = [(H-h)K'+1, \ldots, (H-h+1)K']$, we define the filtration

$$\mathcal{F}_h^k = \sigma\left(\{(s_{h'}^t, a_{h'}^t, r_{h'}^t)\}_{h'\in[H]}^{t\leq k} \cup \{(s_{h'}^{k+1}, a_{h'}^{k+1}, r_{h'}^{k+1})\}_{h'\leq h-1} \cup \{(s_h^{k+1}, a_h^{k+1})\}\right)$$

which is simply all the data up to episode $k+1$ and timestep $h$ but right before $r_h^{k+1}$ and $s_{h+1}^{k+1}$ are generated (in the offline data). Note that for any $k \in \mathcal{I}_h$, we have $(s_h^k, a_h^k, r_h^k, s_{h+1}^k) \in \mathcal{F}_h^k$, and

$$\tilde{V}_{h+1} \in \sigma\left(\{(s_{h'}^k, a_{h'}^k, r_{h'}^k)\}_{h'\in[h+1,\ldots,H]}^{k\in\mathcal{I}_{h'}}\right) \subseteq \mathcal{F}_h^{k-1} \subseteq \mathcal{F}_h^k.$$

Thus, for any $k \in \mathcal{I}_h$, we have

$$\epsilon_h^k = (\mathbb{B}_h\tilde{V}_{h+1})(x_h^k) - r_h^k - \tilde{V}_{h+1}(s_{h+1}^k) \in \mathcal{F}_h^k.$$

Recalling our data splitting strategy $i \in \mathcal{I}_h := [(H-h)K'+1, \ldots, (H-h+1)K']$ with $K' := \lfloor K/H \rfloor$, the key property in our data splitting is that

$$\tilde{V}_{h+1} \in \sigma\left(\{(s_{h'}^k, a_{h'}^k, r_{h'}^k)\}_{h'\in[h+1,\ldots,H]}^{k\in\mathcal{I}_{h'}}\right) \subseteq \mathcal{F}_h^{k-1}.$$

Thus, conditioned on $\mathcal{F}_h^{k-1}$, $\tilde{V}_{h+1}$ becomes deterministic. This implies that

$$\mathbb{E}\left[\epsilon_h^k|\mathcal{F}_h^{k-1}\right] = \left[(\mathbb{B}_h\tilde{V}_{h+1})(s_h^k, a_h^k) - r_h^k - \tilde{V}_{h+1}(s_{h+1}^k)|\mathcal{F}_h^{k-1}\right] = 0.$$

Therefore, for any $h \in [H]$, $\{\epsilon_h^k\}_{k\in\mathcal{I}_h}$ is adapted to the filtration $\{\mathcal{F}_h^k\}_{k\in\mathcal{I}_h}$. Applying Lemma A.4 with $Z_t = \epsilon_t^h \in [-H(1+\iota), H(1+\iota)]$, $\sigma^2 = H^2(1+\iota)^2$, $\rho = \lambda - 1$, for any $\delta > 0$, with probability at least $1 - \delta/3$, for any $h \in [H]$,

$$E_h^T\left[(\mathcal{K}_h + (\lambda - 1)I_{K'})^{-1} + I\right]^{-1}E_h \leq H^2(1+\iota)^2\text{logdet}(\lambda I_{K'} + \mathcal{K}_h) + 2H^2(1+\iota)^2\ln(3H/\delta) \tag{15}$$

Substituting Equation (15) into Equation (14), we have

$$I_5 \leq H^2(1+\iota)^2\text{logdet}(\lambda I_{K'} + \mathcal{K}_h) + 2H^2(1+\iota)^2\ln(3H/\delta)$$
$$= H^2(1+\iota)^2\text{logdet}(I_{K'} + \mathcal{K}_h/\lambda) + H^2(1+\iota)^2K'\ln\lambda + 2H^2(1+\iota)^2\ln(3H/\delta)$$
$$= H^2(1+\iota)^2\tilde{d}_h\ln(1 + K'/\lambda) + H^2(1+\iota)^2K'\ln\lambda + 2H^2(1+\iota)^2\ln(3H/\delta), \tag{16}$$

where the last equation holds due to the definition of the effective dimension.

All together, under the joint event $E\{\text{Lemma C.9}\} \cap E\{\text{Lemma A.1}\} \cap E\{\text{Equation (10)}\} \cap E\{\text{Equation (15)}\}$, with the choice that

$$\lambda > 1, m = \Omega\left(K'^{10}H^8(1+\iota)^8\ln(3K'H/\delta)\right),$$

for any $x \in \mathbb{S}_{d-1}$ and any $h \in [H]$, we have

$$|(\mathbb{B}_h \widetilde{V}_{h+1})(x) - \langle g(x; \theta_0), \hat{\theta}_h^{lin} - \theta_0 \rangle| \leq \frac{B}{\sqrt{m}}(2\sqrt{d} + \sqrt{2\ln(3H/\delta)}) + \xi_h + \gamma \|g(x; \theta_0)\|_{\bar{\Lambda}_h^{-1}},$$

where

$$\gamma = \lambda B \lambda^{-1/2} + \sqrt{K'} C_g^2 \lambda^{-1} (\frac{B}{\sqrt{m}}(2\sqrt{d} + \sqrt{2\ln(3H/\delta)}) + \xi_h)$$

$$+ H(1 + \iota)\sqrt{\tilde{d}_h \ln(1 + K'/\lambda) + K' \ln \lambda + 2\ln(3H/\delta)} + 1$$

by the union bound, this joint event occurs with probability at least $1 - m^{-2} - \delta$.

$\square$

The following lemma characterizes the approximation error between the Bellman target $\bar{Q}_h(x)$ and the functions in the RKHS.

**Lemma C.9.** *Under Assumption 4.1, with probability at least $1 - \delta$ over $w_1, \ldots, w_m$ drawn i.i.d. from $\mathcal{N}(0, I_d/d)$, for any $h \in [H]$, there exist $c_1, \ldots, c_m$ where $c_i \in \mathbb{R}^d$ and $\|c_i\|_2 \leq \frac{B}{m}$ such that*

$$\bar{Q}_h(x) := \sum_{i=1}^m c_i^T x \mathbb{1}\{w_i^T x \geq 0\},$$

$$\|\mathbb{B}_h \tilde{V}_{h+1} - \bar{Q}_h\|_\infty \leq \frac{B}{\sqrt{m}}(2\sqrt{d} + \sqrt{2\ln(H/\delta)}) + \xi_h.$$

*Moreover, $\bar{Q}_h(x)$ can be re-written as*

$$\bar{Q}_h(x) = \langle g(x; \theta_0), \theta_h^* - \theta_0 \rangle$$

*where*

$$\theta_h^* - \theta_0 := \sqrt{m}[a_1 c_1^T, \ldots, a_m c_m^T]^T \in \mathbb{R}^{md}, \text{ and } \|\theta_h^* - \theta_0\|_2 \leq B. \tag{17}$$

*Proof of Lemma C.9.* Let $V_\perp = \arg\inf_{V \in \mathcal{Q}^*} \|V - \mathbb{B}_h \tilde{V}_{h+1}\|_\infty$. We have

$$V_\perp(x) = \int_{\mathbb{R}^d} c(w)^T x \mathbb{1}\{w^T x \geq 0\} dw,$$

for some $c : \mathbb{R}^d \to \mathbb{R}^d$ such that $\sup_w \frac{\|c(w)\|_2}{p_0(w)} \leq B$. By approximation by finite sum in [GCL$^+$19], with probability at least $1 - \delta$, there exist $c_1, \ldots, c_m$ where $c_i \in \mathbb{R}^d$ and $\|c_i\|_2 \leq \frac{B}{m}$ such that

$$\|V_\perp - \bar{Q}_h\|_\infty \leq \frac{B}{\sqrt{m}}(2\sqrt{d} + \sqrt{2\ln(H/\delta)}),$$

where

$$\bar{Q}_h(x) := \sum_{i=1}^m c_i^T x \mathbb{1}\{w_i^T x \geq 0\}.$$

By Assumption 4.1, we have

$$\|\mathbb{B}_h \tilde{V}_{h+1} - \bar{Q}_h\|_\infty \leq \|\mathbb{B}_h \tilde{V}_{h+1} - V_\perp\|_\infty + \|V_\perp - \bar{Q}_h\|_\infty \leq \frac{B}{\sqrt{m}}(2\sqrt{d} + \sqrt{2\ln(H/\delta)}) + \xi_h.$$

$\square$

## C.5 BOUNDS ON THE EMPIRICAL SQUARED BELLMAN RESIDUALS

We bound the empirical squared Bellman residuals $err_h(x_h^k)$ in the following lemma.

**Lemma C.10.** *Let*

$$m = \Omega\left(d^{3/2} R^{-1} \ln^{3/2}(\sqrt{m}/R)\right),$$

$$R = \mathcal{O}\left(m^{1/2} \ln^{-3} m\right),$$

$$(\eta_1 \lambda)^{-1} R^{8/3} m^{-1/3} \ln m \lesssim 1,$$

$$\eta_1 (K' C_g^2 + \lambda/2) \le 1/2,$$

$$T_2 \ge \frac{\ln 2}{\ln(1/(1 - 2\eta\lambda))},$$

$$\sqrt{\tau} \ge 2\gamma,$$

$$M = \ln\frac{H|\mathcal{S}|}{\delta} \Big/ \ln\frac{1}{1 - \Phi(-1)},$$

$$\lambda > 1$$

$$m = \Omega\left(K'^{10} H^8 (1 + \iota)^8 \ln(3K'H/\delta)\right),$$

$$\lambda^{-1}(K'(H - h + 1)^2 (1 + \iota)^2 + 1)\sqrt{K'} R^{1/3} m^{-1/6} \sqrt{m}$$

$$+ \lambda^{-1} K' C_g R^{4/3} m^{-1/6} \sqrt{\ln m} + \lambda^{-1} \sqrt{K'} H(1 + \iota) \lesssim R,$$

*where $\gamma$ is defined in [Lemma C.8](). Set*

$$\iota = R^{4/3} m^{-1/6} \sqrt{\ln m} + \lambda^{-1}(K'H^2(1 + \iota)^2 + 1)\sqrt{K'} R^{1/3} m^{-1/6} \sqrt{m} + \lambda^{-1} K R^{4/3} m^{-1/6} \sqrt{\ln m}$$

$$+ (1 - 2\eta\lambda_{\max}(\bar{\Lambda}_h))^{T_1} \lambda^{-1/2} K' H(1 + \iota) + \gamma\lambda^{-1} \sqrt{K'} R^{1/6} m^{-1/12} \ln^{1/4} m + \gamma\lambda^{-1/2} R^{1/3} m^{-1/6} \sqrt{\ln m}.$$

*With probability at least $1 - m^{-2} - 3\delta$, for any $k, h \in \mathcal{I}_h \times [H]$, we have*

$$err_h(x_h^k) \lesssim \frac{B}{\sqrt{m}}(2\sqrt{d} + \sqrt{2\ln(3H/\delta)}) + \xi_h + \gamma\|g(x_h^k; \theta_0)\|_{\bar{\Lambda}_h^{-1}} + R^{4/3} m^{-1/6} \sqrt{\ln m}$$

$$+ \lambda^{-1}(K'H^2(1 + \iota)^2 + 1)\sqrt{K} R^{1/3} m^{-1/6} \sqrt{m} + \lambda^{-1} K' R^{4/3} m^{-1/6} \sqrt{\ln m}$$

$$+ (1 - 2\eta\lambda_{\max}(\bar{\Lambda}_h))^{T_1} \lambda^{-1/2} K' H(1 + \iota) + \sqrt{2\tau\ln(MK'H/\delta)} \cdot \|g(x_h^k; \theta_0)\|_{\bar{\Lambda}_h^{-1}}$$

$$+ \sqrt{2\tau\ln(MK'H/\delta)} \cdot \left(\lambda^{-1} \sqrt{K'} R^{1/6} m^{-1/12} \ln^{1/4} m + \lambda^{-1/2} R^{1/3} m^{-1/6} \sqrt{\ln m}\right).$$

*Proof of [Lemma C.10]().* Define the event

$$E_1 = \left\{\langle g(x_h^k; \theta_h), \theta_h^{lin,i} - \theta_h^{lin}\rangle \le \sqrt{2\ln(MK'H/\delta)}\|g(x_h^k)\|_{\Sigma_h} : \forall(i, k, h) \in [M] \times \mathcal{I}_h \times [H]\right\}.$$

By [Lemma C.4](), [Lemma A.7]() and the union bound, we have $P(E_1) \ge 1 - \delta$. Now consider the joint event

$$E = E_1 \cap E\{\text{Lemma C.3}\} \cap E\{\text{Lemma C.5}\} \cap E\{\text{Lemma C.6}\} \cap E\{\text{Lemma C.8}\}.$$

The rest of the proof considers under the joint event $E$. Let us define

$$\zeta := R^{4/3} m^{-1/6} \sqrt{\ln m} + \lambda^{-1}(K'H^2(1 + \iota)^2 + 1)\sqrt{K'} R^{1/3} m^{-1/6} \sqrt{m} + \lambda^{-1} K R^{4/3} m^{-1/6} \sqrt{\ln m}$$

$$+ (1 - 2\eta_1\lambda_{\max}(\bar{\Lambda}_h))^{T_1} \lambda^{-1/2} K' H(1 + \iota) + \gamma\lambda^{-1} \sqrt{K'} R^{1/6} m^{-1/12} \ln^{1/4} m + \gamma\lambda^{-1/2} R^{1/3} m^{-1/6} \sqrt{\ln m}.$$

It follows from [Lemma C.5]() that

$$\widetilde{Q}_h(x) = \mathbb{B}_h \widetilde{V}_{h+1}(x) - err_h(x) \le H - h + 1 + (H - h)\iota + \zeta$$
$$\le H - h + 1 + (H - h + 1)\iota,$$

if we choose $\iota \geq \zeta$.

Thus, we have

$$\widetilde{Q}_h(x) = \min\{f(x; \theta_h) + \min_{i \in [M]} \langle g(x; \theta_h), \theta_h^{lin,i} - \theta_h^{lin} \rangle, (H - h + 1)(1 + \iota)\}^+$$
$$= \max\{f(x; \theta_h) + \min_{i \in [M]} \langle g(x; \theta_h), \theta_h^{lin,i} - \theta_h^{lin} \rangle, 0\}.$$

Therefore, for any $(k, h) \in \mathcal{I}_h \times [H]$, we have

$$\mathrm{err}_h(x_h^k) = \mathbb{B}_h \widetilde{V}_{h+1}(x_h^k) - \widetilde{Q}_h(x_h^k)$$
$$= \mathbb{B}_h \widetilde{V}_{h+1}(x_h^k) - f(x_h^k; \theta_h) - \min_{i \in [M]} \langle g(x_h^k; \theta_h), \theta_h^{lin,i} - \theta_h^{lin} \rangle$$
$$\leq \mathbb{B}_h \widetilde{V}_{h+1}(x) - f(x; \theta_h) + \sqrt{2 \ln(MK'H/\delta)} \cdot \|g(x_h^k; \theta_h)\|_{\Sigma_h}$$
$$= \mathbb{B}_h \widetilde{V}_{h+1}(x_h^k) - \langle g(x_h^k; \theta_0), \hat{\bar{\theta}}_h^{lin} - \theta_0 \rangle + \langle g(x_h^k; \theta_0), \hat{\bar{\theta}}_h^{lin} - \theta_0 \rangle - f(x_h^k; \theta_h)$$
$$+ \sqrt{2 \ln(MK'H/\delta)} \cdot \|g(x_h^k; \theta_h)\|_{\Sigma_h}$$
$$\leq \mathbb{B}_h \widetilde{V}_{h+1}(x_h^k) - \langle g(x_h^k; \theta_0), \hat{\bar{\theta}}_h^{lin} - \theta_0 \rangle + \langle g(x_h^k; \theta_0), \hat{\bar{\theta}}_h^{lin} - \theta_0 \rangle - f(x_h^k; \theta_h)$$
$$+ \sqrt{2\tau \ln(MK'H/\delta)} \cdot \|g(x_h^k; \theta_h)\|_{\Lambda_h^{-1}}$$
$$= \underbrace{\mathbb{B}_h \widetilde{V}_{h+1}(x_h^k) - \langle g(x_h^k; \theta_0), \hat{\bar{\theta}}_h^{lin} - \theta_0 \rangle}_{\text{Lemma C.8}} + \underbrace{\langle g(x_h^k; \theta_0), \hat{\bar{\theta}}_h^{lin} - \theta_0 \rangle - f(x_h^k; \theta_h)}_{\text{approx + opt error, Lemma C.6}}$$
$$+ \sqrt{2\tau \ln(MK'H/\delta)} \cdot \|g(x_h^k; \theta_0)\|_{\bar{\Lambda}_h^{-1}}$$
$$+ \sqrt{2\tau \ln(MK'H/\delta)} \cdot \underbrace{\left( \|g(x_h^k; \theta_h)\|_{\Lambda_h^{-1}} - \|g(x_h^k; \theta_0)\|_{\bar{\Lambda}_h^{-1}} \right)}_{\text{Lemma C.3}}$$

where the first inequality holds due event $E_1$, and the second inequality holds due to Equation (8).

$\square$

The following lemma is the NTK analogue of the elliptical potential lemma in [AYPS11].

**Lemma C.11.** *If $\lambda \geq 1$ and $m = \Omega(K'^4 \ln(K'H/\delta))$, then with probability at least $1 - \delta$, for any $h \in [H]$, we have*

$$\sum_{k \in \mathcal{I}_h} \|g(x_h^k; \theta_0)\|_{\bar{\Lambda}_h^{-1}}^2 \leq 2\tilde{d}_h \ln(1 + K'/\lambda) + 1.$$

*Proof of Lemma C.11.* Define

$$\bar{\Lambda}_h^k := \lambda I + \mathbb{1}\{k \in \mathcal{I}_h\} \sum_{i=1}^{k-1} g(x_h^i; \theta_0) g(x_h^i; \theta_0)^T.$$

Then we have $\bar{\Lambda}_h^{-1} \preceq (\bar{\Lambda}_h^k)^{-1}, \forall k \in [K]$. Thus, we have

$$\sum_{k \in \mathcal{I}_h} \|g(x_h^k; \theta_0)\|_{\bar{\Lambda}_h^{-1}}^2 \leq \sum_{k \in \mathcal{I}_h} \|g(x_h^k; \theta_0)\|_{(\bar{\Lambda}_h^k)^{-1}}^2.$$

For any fixed $h \in [H]$, let

$$U = [g(x_h^k; \theta_0)]_{k \in \mathcal{I}_h} \in \mathbb{R}^{md \times K'}.$$

By the union bound, with probability at least $1 - \delta$, for any $h \in [H]$, we have

$$\sum_{k \in \mathcal{I}_h} \|g(x_h; \theta_0)\|_{(\Lambda_h^k)^{-1}}^2 \leq 2 \ln \frac{\det \bar{\Lambda}_h}{\det(\lambda I)}$$

$$= 2\text{logdet}\left(I + \sum_{k \in \mathcal{I}_h} g(x_h^k; \theta_0)g(x_h^k; \theta_0)^T/\lambda\right)$$

$$= 2\text{logdet}(I + UU^T/\lambda)$$

$$= 2\text{logdet}(I + U^TU/\lambda)$$

$$= 2\text{logdet}(I + \mathcal{K}_h/\lambda + (U^TU - \mathcal{K}_h)/\lambda)$$

$$\leq 2\text{logdet}(I + \mathcal{K}_h/\lambda) + 2\text{tr}\left((I + \mathcal{K}_h/\lambda)^{-1}(U^TU - \mathcal{K}_h)/\lambda\right)$$

$$\leq 2\text{logdet}(I + \mathcal{K}_h/\lambda) + 2\|(I + \mathcal{K}_h/\lambda)^{-1}\|_F\|U^TU - \mathcal{K}_h\|_F$$

$$\leq 2\text{logdet}(I + \mathcal{K}_h/\lambda) + 2\sqrt{K'}\|U^TU - \mathcal{K}_h\|_F$$

$$\leq 2\text{logdet}(I + \mathcal{K}_h/\lambda) + 1$$

$$= 2\tilde{d}_h \ln(1 + K'/\lambda) + 1$$

where the first inequality holds due to $\lambda \geq C_g^2$ and [AYPS11, Lemma 11], the third equality holds due to that $\text{logdet}(I + AA^T) = \text{logdet}(I + A^TA)$, the second inequality holds due to that $\text{logdet}(A + B) \leq \text{logdet}(A) + \text{tr}(A^{-1}B)$ as the result of the convexity of $\text{logdet}$, the third inequality holds due to that $\text{tr}(A) \leq \|A\|_F$, the fourth inequality holds due to $2\sqrt{K'}\|U^TU - \mathcal{K}_h\|_F \leq 1$ by the choice of $m = \Omega(K'^4 \ln(K'H/\delta))$, Lemma A.2 and the union bound, and the last equality holds due to the definition of $\tilde{d}_h$. □

## C.6 PROOF OF THEOREM 2

We are now ready to present the proof of Theorem 2.

*Proof of Theorem 2.* We start with the value difference lemma [JYW21]: For any policy $\pi$ (including stochastic and non-Markovian policies), we have

$$V_1^\pi(s_1) - V_1^{\tilde{\pi}}(s_1) = \sum_{h=1}^H \mathbb{E}_\pi\left[\text{err}_h(s_h, a_h)\right] - \sum_{h=1}^H \mathbb{E}_{\tilde{\pi}}\left[\text{err}_h(s_h, a_h)\right]$$

$$+ \sum_{h=1}^H \mathbb{E}_\pi \underbrace{\left[\langle \widetilde{Q}_h(s_h, \cdot), \pi_h(\cdot|s_h) - \widetilde{\pi}_h(\cdot|s_h)\rangle\right]}_{\leq 0}$$

$$\leq \sum_{h=1}^H \mathbb{E}_\pi\left[\text{err}_h(s_h, a_h)\right] - \sum_{h=1}^H \mathbb{E}_{\tilde{\pi}}\left[\text{err}_h(s_h, a_h)\right],$$

where the inequality follows from that $\widetilde{\pi}$ is greedy with respect to $\widehat{Q}_h$. To bound the first term, we use Lemma A.3, Lemma C.10, and Lemma C.11. The second term is bounded using Lemma C.5.

We now give the characterization of the hyperparameters in Neural-LMC-PPS that arise in Theorem 2. All together the parameter conditions of Lemma A.3, Lemma C.10, and Lemma C.11, the parameter conditions are:

$$m = \Omega\left(d^{3/2}R^{-1}\ln^{3/2}(\sqrt{m}/R)\right),$$

$$R = \mathcal{O}\left(m^{1/2}\ln^{-3} m\right),$$

$$(\eta_1\lambda)^{-1}R^{8/3}m^{-1/3}\ln m \lesssim 1,$$

$$\eta_1(K'C_g^2 + \lambda/2) \leq 1/2,$$

$$T_2 \geq \frac{\ln 2}{\ln(1/(1 - 2\eta\lambda))},$$

$$\sqrt{\tau} \geq 2\gamma,$$

$$M = \ln\frac{HS}{\delta}\Big/\ln\frac{1}{1 - \Phi(-1)},$$

$$\lambda > 1,$$
$$m = \Omega\left(K'^{10}H^8(1+\iota)^8\ln(3K'H/\delta)\right),$$
$$M = \ln\frac{H|\mathcal{S}|}{\delta}\Big/\ln\frac{1}{1-\Phi(-1)},$$
$$\lambda^{-1}(K'(H-h+1)^2(1+\iota)^2+1)\sqrt{K'}R^{1/3}m^{-1/6}\sqrt{m}$$
$$+\lambda^{-1}K'C_gR^{4/3}m^{-1/6}\sqrt{\ln m}+\lambda^{-1}\sqrt{K'}H(1+\iota)\lesssim R,$$

where

$$\gamma \lesssim B\sqrt{\lambda}+\sqrt{K'}\lambda^{-1}\left(\frac{B}{\sqrt{m}}(2\sqrt{d}+\sqrt{2\ln(3H/\delta)})+\xi_h\right)$$

$$+ H(1+\iota)\sqrt{\tilde{d}_h\ln(1+K'/\lambda)+K'\ln\lambda+2\ln(3H/\delta)}+1,$$
$$\iota = R^{4/3}m^{-1/6}\sqrt{\ln m}+\lambda^{-1}(K'H^2(1+\iota)^2+1)\sqrt{K'}R^{1/3}m^{-1/6}\sqrt{m}+\lambda^{-1}KR^{4/3}m^{-1/6}\sqrt{\ln m}$$
$$+ (1-2\eta\lambda_{\max}(\bar{\Lambda}_h))^{T_1}\lambda^{-1/2}K'H(1+\iota)+\gamma\lambda^{-1}\sqrt{K'}R^{1/6}m^{-1/12}\ln^{1/4}m$$
$$+ \gamma\lambda^{-1/2}R^{1/3}m^{-1/6}\sqrt{\ln m}.$$

It is easy to see that there exists $m \gtrsim \text{poly}(K', H, d, B, \lambda, 1/\delta)$, $R = \Omega(H\sqrt{K'})$ and sub-polynomial in $m$ that satisfy the parameter conditions above.

$\square$

## APPENDIX D   EXPERIMENT DETAILS

In this section, we give more details of our experiments in Section 5.

### D.1   ALGORITHM DETAILS

We give the detailed accounts of all algorithms we used in our experiment in Section 5: LinLCB in Algorithm 4, NeuraLCB in Algorithm 5, NeuralGreedy in Algorithm 6, Neural-LMC-PPS(simplified) in Algorithm 7, and NeuralTS in Algorithm 8.

Several remarks are in order. Neural-LMC-PPS(simplified) in Algorithm 7 is a simplified version of Neural-LMC-PPS, where the former directly applies Langevin dynamics to obtain approximate posterior weight samples, without using auxiliary linear models like the original Neural-LMC-PPS. NeuralTS in Algorithm 8 simply perturbs the value predictor $f(\cdot,\cdot;\theta_h^i)$ by an amount of $\epsilon_i\|g(\cdot,\cdot;\theta_i)\|_{\Lambda_h^{-1}}$ that is scaled with the weighted norm of the network gradient. NeuraLCB in Algorithm 5 modifies the original NeuraLCB in [NTGNV22] where we use gradient descent instead of stochastic gradient descent.

---

**Algorithm 4** LinLCB/PEVI [JYW21]

---

**Input:** Offline data $\mathcal{D} = \{(s_h^k, a_h^k, r_h^k)\}_{h\in[H]}^{k\in[K]}$, uncertainty multiplier $\beta$, regularization parameter $\lambda$.

1: Initialize $\tilde{V}_{H+1}(\cdot) \leftarrow 0$
2: **for** $h = H, \ldots, 1$ **do**
3:    $\quad \Lambda_h \leftarrow \sum_{k=1}^{K}\phi(s_h^k, a_h^k)\phi(s_h^k, a_h^k)^T + \lambda I$
4:    $\quad \hat{\theta}_h \leftarrow \Sigma_h^{-1}\sum_{k=1}^{K}\phi_h(s_h^k, a_h^k)\cdot(r_h^k + \hat{V}_{h+1}(s_{h+1}^k))$
5:    $\quad b_h(\cdot,\cdot) \leftarrow \beta\cdot\|\phi_h(\cdot,\cdot)\|_{\Sigma_h^{-1}}.$
6:    $\quad \hat{Q}_h(\cdot,\cdot) \leftarrow \min\{\langle\phi_h(\cdot,\cdot),\hat{\theta}_h\rangle - b_h(\cdot,\cdot), H-h+1\}^+.$
7:    $\quad \hat{\pi}_h \leftarrow \arg\max_{\pi_h}\langle\hat{Q}_h, \pi_h\rangle$ and $\hat{V}_h^k \leftarrow \langle\hat{Q}_h^k, \pi_h^k\rangle.$
8: **end for**
**Output:** $\hat{\pi} = \{\hat{\pi}_h\}_{h\in[H]}$

---

---

**Algorithm 5** NeuraLCB (a modification of [NTGNV22])

---

**Input:** Offline data $\mathcal{D} = \{(s_h^k, a_h^k, r_h^k)\}_{h \in [H]}^{k \in [K]}$, neural networks $\{f(\cdot, \cdot; \theta) : \theta \in \Theta\}$, uncertainty multiplier $\beta$, regularization parameter $\lambda$, step size $\eta$, number of gradient descent steps $J$

1: Initialize $\tilde{V}_{H+1}(\cdot) \leftarrow 0$ and initialize $f(\cdot, \cdot; W)$ with initial parameter $W_0$
2: **for** $h = H, \ldots, 1$ **do**
3: $\quad \hat{W}_h \leftarrow \text{GD}(\lambda, \eta, J, \{(s_h^k, a_h^k, r_h^k)\}_{k \in [K]}, 0, W_0)$ (Algorithm 10)
4: $\quad \Lambda_h = \lambda I + \sum_{k=1}^K g(s_h^k, a_h^k; \hat{W}_h) g(x_h^k; \hat{W}_h)^T$
5: $\quad$ Compute $\hat{Q}_h(\cdot, \cdot) \leftarrow \min\{f(\cdot, \cdot; \hat{W}_h) - \beta \|g(\cdot, \cdot; \hat{W}_h)\|_{\Lambda_h^{-1}}, H - h + 1\}^+$
6: $\quad \hat{\pi}_h \leftarrow \arg\max_{\pi_h} \langle \hat{Q}_h, \pi_h \rangle$ and $\hat{V}_h \leftarrow \langle \hat{Q}_h, \hat{\pi}_h \rangle$
7: **end for**
**Output:** $\hat{\pi} = \{\hat{\pi}_h\}_{h \in [H]}$.

---

**Algorithm 6** NeuralGreedy

---

**Input:** Offline data $\mathcal{D} = \{(s_h^k, a_h^k, r_h^k)\}_{h \in [H]}^{k \in [K]}$, neural networks $\{f(\cdot, \cdot; \theta) : \theta \in \Theta\}$, step size $\eta$, number of gradient descent steps $T$

1: Initialize $\tilde{V}_{H+1}(\cdot) \leftarrow 0$ and initialize $f(\cdot, \cdot; W)$ with initial parameter $W_0$
2: **for** $h = H, \ldots, 1$ **do**
3: $\quad \theta_h \leftarrow GD(\mathcal{L}_h, \theta_0, \eta, T)$ (Algorithm 10) where $\mathcal{L}_h$ is defined in Line 2 of Algorithm 3.
4: $\quad$ Compute $\hat{Q}_h(\cdot, \cdot) \leftarrow \min\{f(\cdot, \cdot; \theta_h), H - h + 1\}^+$
5: $\quad \hat{\pi}_h \leftarrow \arg\max_{\pi_h} \langle \hat{Q}_h, \pi_h \rangle$ and $\hat{V}_h \leftarrow \langle \hat{Q}_h, \hat{\pi}_h \rangle$
6: **end for**
**Output:** $\hat{\pi} = \{\hat{\pi}_h\}_{h \in [H]}$.

---

**Algorithm 9** GLD$(L(\theta), \theta_0, \eta, T, \tau)$: Gradient Langevin dynamics

---

1: **for** $t = 1 \ldots T$ **do**
2: $\quad \theta_t \leftarrow \theta_{t-1} - \eta \nabla_\theta L(\theta_{t-1}) + \sqrt{2\eta\tau} \epsilon_t$ where $\epsilon_t \sim \mathcal{N}(0, I)$
3: **end for**
**Output:** $\theta_T$

---

**Algorithm 10** GD$(L(\theta), \theta_0, \eta, T)$: Gradient descent

---

1: **for** $t = 1 \ldots T$ **do**
2: $\quad \theta_t \leftarrow \theta_{t-1} - \eta \nabla_\theta L(\theta_{t-1})$
3: **end for**
**Output:** $\theta_T$

---

### D.2 EXPERIMENTAL SETUP AND TRAINING DETAILS

We give the details of our experimental setup and training of the empirical results presented in Section 5.

### LINEAR MDPs

In this appendix, we provide further details on the experiment setup. We describe in detail a variant of the hard instance of linear MDPs [MWZG21] used in our experiment. The linear MDP has $\mathcal{S} = \{0, 1\}$, $\mathcal{A} = \{0, 1, \cdots, 99\}$, and the feature dimension $d = 10$. Each action $a \in [99] = \{1, \ldots, 99\}$ is represented by its binary encoding vector $u_a \in \mathbb{R}^8$ with entry being either $-1$ or $1$. The feature mapping $\phi(s, a)$ is given by $\phi(s, a) = [u_a^T, \delta(s, a), 1 - \delta(s, a)]^T \in \mathbb{R}^{10}$, where $\delta(s, a) = 1$ if $(s, a) = (0, 0)$ and $\delta(s, a) = 0$ otherwise. The true measure $\nu_h(s)$ is given by $\nu_h(s) = [0, \cdots, 0, (1 - s) \oplus \alpha_h, s \oplus \alpha_h]$ where $\{\alpha_h\}_{h \in [H]} \in \{0, 1\}^H$ are generated uniformly at random and $\oplus$ is the XOR operator. We define $\theta_h = [0, \cdots, 0, r, 1 - r]^T \in \mathbb{R}^{10}$ where $r = 0.99$.

---

**Algorithm 7** Neural-LMC-PPS(simplified)

---

**Input:** Dataset $\mathcal{D} = \{(s_h^k, a_h^k, r_h^k)\}_{h \in [H]}^{k \in [K]}$, neural networks $\{f(\cdot, \cdot; \theta) : \theta \in \Theta\}$, step size $\eta$, temperature parameter $\tau$, regularization parameter $\lambda$, number of training iterations $T$, ensemble size $M$, clipping factor $\iota$
1: Initialize $\widetilde{V}_{H+1}(\cdot) \leftarrow 0$ and initialize $\theta_0$
2: **for** step $h = H, H-1, \ldots, 1$ **do**
3:      **for** $i = 1 \ldots M$ **do**
4:          $\theta_h^i \leftarrow GLD(\mathcal{L}_h, \theta_0, \eta, T, \tau)$ (Algorithm 9) where $\mathcal{L}_h$ is defined in Line 2 of Algorithm 3.
5:          $\tilde{f}_h^i(\cdot, \cdot) \leftarrow f(\cdot, \cdot; \theta_h^i)$
6:      **end for**
7:      $\widetilde{Q}_h(\cdot, \cdot) \leftarrow \min\{\min_{i \in [M]} \tilde{f}_h^i(\cdot, \cdot), (H-h+1)(1+\iota)\}^+$
8:      $\widetilde{\pi}_h \leftarrow \arg\max_{\pi_h \in \Pi} \langle \widetilde{Q}_h, \pi_h \rangle$
9:      $\widetilde{V}_h(\cdot) \leftarrow \langle \widetilde{Q}_h(\cdot, \cdot), \widetilde{\pi}_h(\cdot|\cdot) \rangle$.
10: **end for**
**Output:** $\widetilde{\pi} = \{\widetilde{\pi}_h\}_{h \in [H]}$.

---

**Algorithm 8** NeuralTS

---

**Input:** Dataset $\mathcal{D} = \{(s_h^k, a_h^k, r_h^k)\}_{h \in [H]}^{k \in [K]}$, neural networks $\{f(\cdot, \cdot; \theta) : \theta \in \Theta\}$, step size $\eta$, temperature parameter $\tau$, regularization parameter $\lambda$, number of training iterations $T$, ensemble size $M$, clipping factor $\iota$.
1: Initialize $\widetilde{V}_{H+1}(\cdot) \leftarrow 0$ and initialize $\theta_0$
2: **for** step $h = H, H-1, \ldots, 1$ **do**
3:      **for** $i = 1 \ldots M$ **do**
4:          $\theta_h^i \leftarrow GD(\mathcal{L}_h, \theta_0, \eta, T)$ (Algorithm 10) where $\mathcal{L}_h$ is defined in Line 2 of Algorithm 3.
5:          Draw $\epsilon_i \sim \mathcal{N}(0, \sigma^2)$
6:          $\tilde{f}_h^i(\cdot, \cdot) \leftarrow f(\cdot, \cdot; \theta_h^i) + \epsilon_i \|g(\cdot, \cdot; \theta_i)\|_{\Lambda_h^{-1}}$ where $\Lambda_h := \lambda I + \sum_{k=1}^{K} g(x_h^k; \theta_h) g(x_h^k; \theta_h)^T$
7:      **end for**
8:      $\widetilde{Q}_h(\cdot, \cdot) \leftarrow \min\{\min_{i \in [M]} \tilde{f}_h^i(\cdot, \cdot), (H-h+1)(1+\iota)\}^+$
9:      $\widetilde{\pi}_h \leftarrow \arg\max_{\pi_h \in \Pi} \langle \widetilde{Q}_h, \pi_h \rangle$
10:      $\widetilde{V}_h(\cdot) \leftarrow \langle \widetilde{Q}_h(\cdot, \cdot), \widetilde{\pi}_h(\cdot|\cdot) \rangle$.
11: **end for**
**Output:** $\widetilde{\pi} = \{\widetilde{\pi}_h\}_{h \in [H]}$.

---

Recall that the transition follows $\mathbb{P}_h(s'|s, a) = \langle \phi(s, a), \nu_h(s') \rangle$ and the mean reward $r_h(s, a) = \langle \phi(s, a), \theta_h \rangle$. We generated a priori $K \in \{1, \ldots, 1000\}$ trajectories using the behavior policy $\mu$, where for any $h \in [H]$ we set $\mu_h(0|0) = p, \mu_h(1|0) = 1 - p, \mu_h(a|0) = 0, \forall a > 1; \mu_h(0|1) = p, \mu_h(a|1) = (1-p)/99, \forall a > 0$, where we set $p = 0.6$.

We run over $K \in \{1, \ldots, 1000\}$ and $H \in \{20, 30, 50, 80\}$. We set $\lambda = 0.01$ for all algorithms. For LinPER, we grid searched $\sigma_h = \sigma \in \{0.0, 0.1, 0.5, 1.0, 2.0\}$ and $M \in \{1, 2, 10, 20\}$. For LinLCB, we grid searched its uncertainty multiplier $\beta \in \{0.1, 0.5, 1, 2\}$. The sub-optimality metric is used to compare algorithms. For each $H \in \{20, 30, 50, 80\}$, each algorithm was executed for 30 times, and the averaged results (with std) are reported in Figure 1.

NON-LINEAR CONTEXTUAL BANDITS

In this appendix, we provide in detail the experimental and hyperparameter setup in our experiment. To predict the value of different actions from the same state $s$ using neural networks, we transform a state $s \in \mathbb{R}^d$ into $dA$-dimensional vectors $s^{(1)} = (s, 0, \ldots, 0), s^{(2)} = (0, s, 0, \ldots, 0), \ldots, s^{(A)} = (0, \ldots, 0, s)$ and train the network to map $s^{(a)}$ to $r(s, a)$ given a pair of data $(s, a)$.

For NeuralGreedy, NeuraLCB, NeuralTS and Neural-LMC-PPS, we use the same neural network architecture with two hidden layers whose width $m = 64$, train the network with SGD optimizer

with learning rate being grid-searched over $\{0.001, 0.01, 0.05, 0.1\}$ and batch size of $64$. For NeuraLCB, and LinLCB, we grid-searched $\beta$ over $\{0.001, 0.01, 0.1, 1, 5, 10\}$. For NeuralTS, we grid-searched $\sigma_{TS} \in \{0.001, 0.01, 0.1, 1, 5, 10\}$, and $M \in \{1, 10, 20, 50\}$. For Neural-LMC-PPS, we grid-searched $\tau \in \{0.00001, 0.0001, 0.001, 0.01, 0.1, 1, 5, 10, 100\}$ and $M \in \{1, 10, 20\}$. We fixed the regularization parameter $\lambda = 0.01$ for all algorithms and the offline data is generated by a uniform behavior policy. To estimate the expected sub-optimality, we randomly obtain $1,000$ novel samples (i.e. not used in training) to compute the average sub-optimality and keep these same samples for all algorithms.

