# OpenReview forum: "Posterior Sampling via Langevin Monte Carlo for Offline Reinforcement Learning"
_ICLR.cc/2024/Conference — Submitted to ICLR 2024_

### Official Review · Reviewer_VjxR · 2023-10-29

**Soundness:** 4 excellent
**Presentation:** 4 excellent
**Contribution:** 3 good
**Rating:** 6
**Confidence:** 4

**Summary:**

The authors present a model-free posterior sampling approach for offline RL using Langevin Monte Carlo (LMC) for posterior approximation. They introduce practical algorithms in an episodic setting for both linear low-rank MDPs, and general MDPs (with over-parameterized neural networks for value function approximation, alongside an auxiliary linear model for LMC). Notably, the paper establishes frequentist sub-optimal bounds both cases. Empirical evaluations on linear MDP and non-linear contextual bandits support the proposed algorithms' effectiveness.

**Strengths:**

I believe the most important strength is that the paper offers an insightful advancement in offline RL through a Bayesian lens. While the value-based variation to classical PSRL and the employment of LMC for posterior approximation are not novelties in isolation, their integration within offline RL is both meaningful and aptly executed.

The implicit pessimism by posterior sampling with the proof of a frequentist bound is also a non-trivial contribution, and provides a fresh perspective to ongoing discussions in this domain.

**Weaknesses:**

While the paper makes significant theoretical advancements, it would further solidify its applicability if the proposed algorithms were tested on well-regarded benchmarks, such as the MuJoCo tasks from the D4RL suite. Additional experiments with model-based approaches would offer a comprehensive perspective on the approach's effectiveness.

The presented approach captures pessimism through posterior sampling. While innovative, one could question whether this form of pessimism adequately represents the complex nature of uncertainties found in the offline dataset, particularly given the non-stationary distributions that can arise from varied data collection policies.

**Questions:**

Please refer to the concerns in the weaknesses part.

---

> ### Author Response · Authors · 2023-11-20
>
> We thank the reviewer for acknowledging our contributions and for the insightful questions.
>
> > The presented approach captures pessimism through posterior sampling. While innovative, one could question whether this form of pessimism adequately represents the complex nature of uncertainties found in the offline dataset, particularly given the non-stationary distributions that can arise from varied data collection policies.
>
> This is indeed a very insightful question. The short answer is we do not know yet for sure. We speculate that this form of pessimism in our algorithms is likely not tightly capturing the uncertainties in the offline dataset as there is still a gap of $\sqrt{d}$ of our bounds for linear MDPs in the worst-case scenarios (please see the "Interpolating Bounds" paragraph on page 7). Closing this gap of approximate posterior sampling (perhaps with better algorithms that can tightly capture sufficient pessimism for offline RL, per your suggestion) is left as a future direction.

---

> > ### Author Response · Authors · 2023-11-22
> > **Follow-up questions**
> >
> > Dear reviewer VjxR, thanks again for your supportive comments and insightful questions! Could you let us know if your concerns have been addressed? We would be happy to provide further explanations if you have any.

---

### Official Review · Reviewer_jpdm · 2023-10-30

**Soundness:** 3 good
**Presentation:** 2 fair
**Contribution:** 3 good
**Rating:** 5
**Confidence:** 4

**Summary:**

This paper explores convergence of posterior sampling via Langevin Monte Carlo for offline RL.

**Strengths:**

The study sounds solid, although I did not go through each step of the proof.

**Weaknesses:**

The paper did not clearly explain the fundamental difference between the convergence of Langevin Monte Carlo and the convergence of the RL posterior sampling under the offline setting.

**Questions:**

1. What is the difference between the convergence of Langevin Monte Carlo and the convergence of the RL posterior sampling under the offline setting? Will the former lead to the latter?

2. Why is LMC, instead of SGLD, used in Algorithms 2 and 3?  Can mini-batch data be used in simulations of the proposed algorithm?

---

> ### Author Response · Authors · 2023-11-20
>
> We thank the reviewer for the positive reviews.
>
> ---
> > What is the difference between the convergence of Langevin Monte Carlo and the convergence of the RL posterior sampling under the offline setting? Will the former lead to the latter?
>
> To clarify the context, we assume that you are questioning the relationship between the Langevin Monte Carlo (LMC-PPS) way in Algorithm 2 and the pessimistic posterior sampling way in Algorithm 1. Note standard posterior sampling in [US21] does not have any frequentist guarantee. Please let us know if you are implying other stuff.  We are happy to elaborate on that more.
>
> Responding to the relation between the two algorithms, we discussed this at length in the second paragraph of Section 6. For the reviewer’s convenience, we summarize it here. In our settings, LMC-PPS and exact posterior sampling (PPS with exact posterior samples, not with approximate samples via LMC) achieve the same guarantees. This shows a promising benefit of LMC-PPS, where we can employ first-order sampling methods (e.g., LMC) without compromising the statistical guarantees. This is immensely meaningful as, in many cases, exact posterior sampling is much more expensive to obtain (in our settings, as formally discussed in Section 6) or even intractable, while LMC is simply a noisy gradient-based method that can efficiently apply to any differentiable model.
>
> ---
>
> > Why is LMC, instead of SGLD, used in Algorithms 2 and 3? Can mini-batch data be used in simulations of the proposed algorithm?
>
> We can replace LMC with SGLD, where the use of stochastic gradients introduces additional complexities that are typically managed by the standard tools in LMC [1]. In essence, we only need to account for the additional source of stochasticity stemming from the random batch selection in the analysis. However, it's important to note that this point is ancillary to our main argument. Consequently, we introduce LMC for the sake of clarity in exposition.
>
> In our simulations, we did, in fact, utilize mini-batch data (SGLD). Moreover, we discuss in Section D.2 in our appendix for a thorough description of the experimental setup and training particulars. We will enhance the elucidation of the relationship between LMC and SGLD in the final version.
>
> Reference:
>
> [1] AS Dalalyan, A Karagulyan. User-friendly guarantees for the Langevin Monte Carlo with inaccurate gradient. Stochastic Processes and their Applications, 2019

---

> ### Author Response · Authors · 2023-11-22
> **Follow-up comments**
>
> Dear reviewer jpdm,
>
> As the author-reviewer discussion period will end soon, we will appreciate it if you could check our response to your review comments. We are confident that we have addressed your concern in the rebuttal. We haven't hear from you, could you let us know if your concerns have been addressed? If our response resolves your concerns, we kindly ask you to consider raising the rating of our work. If not, we are very happy to provide further explanations. Thank you very much for your time and efforts!

---

### Official Review · Reviewer_Cabd · 2023-11-01

**Soundness:** 3 good
**Presentation:** 2 fair
**Contribution:** 2 fair
**Rating:** 6
**Confidence:** 3

**Summary:**

The submission studies a Bayesian method for offline RL. The proposed method is quite simple (which I see as a pro), simply do the noisy gradient descent on the regression objective. Analysis with improved bounds is the main contribution of the paper.

**Strengths:**

- The proposed method is simple and seems implementable in practice.

- The bounds are improved from previous work.

- Analysis of the NTK regime is performed, which might be of independent interest.

**Weaknesses:**

- I do not see particularly new ideas from the submission, either in the algorithm or in the analysis. Thus, the novelty of the paper is limited.

- The work in Uehara and Sun [US21] considers the setting where the representation of state-action $\phi(s,a)$ is unknown, whereas the submission assumes the feature representation function is known. I think it is not fair to claim the improvement from [US21].

In general, even though this is a technical paper, the submission is a bit hard to parse.

- How is the regression objective related to posterior sampling? How is this a "Langevin" Monte Carlo method? It would be good to be introductory to Langevin Monte Carlo methods, and how the concepts are attached to the actual algorithm presented.

- I had to understand the linear (or low-rank) MDP part very clearly before paying attention to the NTK function approximation part. I do not see any particular contribution from the NTK part to reinforcement learning theory. It is a good add-on result though.

**Questions:**

- I wonder how this method performs on some offline deep-RL benchmarks.

---

> ### Author Response · Authors · 2023-11-20
>
> We thank the reviewer for the positive reviews and the constructive feedback. We will revise our paper accordingly based on your suggestions.
>
> ---
> > I do not see particularly new ideas from the submission, either in the algorithm or in the analysis. Thus, the novelty of the paper is limited.
>
> For algorithms, Lin-LMC-PPS and Neural-LMC-PPS in our paper are both novel for offline RL. Especially, the use of linear auxiliary models in our Neural-LMC-PPS is non-standard. For analysis, we did not claim to invent any new technical tools and we built upon some of the existing techniques used for linear MDP [JYW21] and Neural MDP [NTA23]  to analyze our problem settings. However, the fact that our algorithms are completely different from their algorithms by nature and that our bounds improve upon the bounds of [JYW21] and [NTA23] in the respective MDP models shows that we cannot use their techniques ``as is'' and that we need a new analysis treatment.  The key technical challenge that is absent from both [JYW21] and [NTA23] is to show whether we can obtain pessimism only from **approximate** posterior samples. To address this challenge (and obtain such improvement consequently), the key idea is to separate the distributional shift problem from the estimation problem (Lemma A.3 in our appendix) and leverage the (near-)linear structures of linear MDP and neural MDPs to control the noises propagated from **approximate** posterior samples.
>
> ---
> > The work in Uehara and Sun [US21] considers the setting where the representation of state-action is unknown, whereas the submission assumes the feature representation function is known. I think it is not fair to claim the improvement from [US21].
>
> We apologize for the confusion.  We compared only with Section 8 of [US21], which studies the model-based Posterior Sampling for offline RL where there is no representation learning (the rest of [US21] is not related to posterior sampling). We will revise our paper accordingly.
>
> ---
> > How is the regression objective related to posterior sampling? How is this a "Langevin" Monte Carlo method?
>
> The regression objective is the negative log-likelihood function of the posterior (see footnote 7 in our paper). If one simply samples from $\theta\sim \exp(-L_h(\theta))$, then it corresponds to the standard posterior sampling (with uninformative prior). However, direct sampling from the general $\exp(-L_h(\theta))$ could be intractable, so the alternative method would be adding noise to the gradient decent step of the negative log-likelihood function, which is essentially Langevin Monte Carlo (also check eqn (4) of [WT11]).
>
> ---
> >  I do not see any particular contribution from the NTK part to reinforcement learning theory. It is a good add-on result though.
>
> We study the statistical benefits of Posterior Sampling via Langevin Monte Carlo for offline RL in neural MDPs using NTK. Linear MDPs and neural MDPs (with NTK) are two different models that cover two different aspects of scenarios we might encounter. The guarantees in linear MDPs do not imply any guarantees in neural MDPs (in fact, roughly speaking, the bound for neural MDPs is worse than that for linear MDP by a factor of $\sqrt{H}$); thus, considering both settings gives two orthogonal views of the landscape of  Posterior Sampling via Langevin Monte Carlo for offline RL.
>
> ---
> > I wonder how this method performs on some offline deep-RL benchmarks.
>
> While the current scope of our paper focuses only on the theoretical understanding of the offline LMC algorithm, the empirical performances of our algorithm in large-scale scenarios remain an interesting and non-trivial question that we leave as future work.

---

> > ### Author Response · Authors · 2023-11-22
> > **Follow-up comments**
> >
> > Dear reviewer Cabd, thanks again for the comments, and we believe we have addressed your concern in the rebuttal. We haven't hear from you, could you let us know if your concerns have been addressed? We would be happy to provide further explanations if you have any.

---

### Meta-Review · Area_Chair_3fk6 · 2023-12-21

**Metareview:**

This paper proposes a practical posterior sampling algorithm via Langevin Monte Carlo for offline reinforcement learning (RL) and derives a (frequentist) sub-optimality bound that competes against any comparator policy. It also showcases its applications in low-rank MDPs and RL with neural network approximation. The main concerns about this paper include: (1) the novelty of the algorithm and analysis is limited;  (2) the use of low-rank MDPs is misleading, the authors study linear MDP rather than low-rank MDPs (unknown feature mapping). Even after the author response, this paper does not gather sufficient support. Therefore, I recommend rejection.

**Justification For Why Not Higher Score:**

Even after the author response, this paper does not gather sufficient support.

**Justification For Why Not Lower Score:**

N/A

---

### Decision · Program_Chairs · 2024-01-16

Reject